# ON THE CONVERGENCE OF ADAM AND BEYOND

**Sashank J. Reddi, Satyen Kale & Sanjiv Kumar**
Google New York
New York, NY 10011, USA
`{sashank,satyenkale,sanjivk}@google.com`

## ABSTRACT

Several recently proposed stochastic optimization methods that have been successfully used in training deep networks such as RMSPROP, ADAM, ADADELTA, NADAM are based on using gradient updates scaled by square roots of exponential moving averages of squared past gradients. In many applications, e.g. learning with large output spaces, it has been empirically observed that these algorithms fail to converge to an optimal solution (or a critical point in nonconvex settings). We show that one cause for such failures is the exponential moving average used in the algorithms. We provide an explicit example of a simple convex optimization setting where ADAM does not converge to the optimal solution, and describe the precise problems with the previous analysis of ADAM algorithm. Our analysis suggests that the convergence issues can be fixed by endowing such algorithms with "long-term memory" of past gradients, and propose new variants of the ADAM algorithm which not only fix the convergence issues but often also lead to improved empirical performance.

## 1 INTRODUCTION

Stochastic gradient descent (SGD) is the dominant method to train deep networks today. This method iteratively updates the parameters of a model by moving them in the direction of the negative gradient of the loss evaluated on a minibatch. In particular, variants of SGD that scale coordinates of the gradient by square roots of some form of averaging of the squared coordinates in the past gradients have been particularly successful, because they automatically adjust the learning rate on a per-feature basis. The first popular algorithm in this line of research is ADAGRAD (Duchi et al., 2011; McMahan & Streeter, 2010), which can achieve significantly better performance compared to vanilla SGD when the gradients are sparse, or in general small.

Although ADAGRAD works well for sparse settings, its performance has been observed to deteriorate in settings where the loss functions are nonconvex and gradients are dense due to rapid decay of the learning rate in these settings since it uses all the past gradients in the update. This problem is especially exacerbated in high dimensional problems arising in deep learning. To tackle this issue, several variants of ADAGRAD, such as RMSPROP (Tieleman & Hinton, 2012), ADAM (Kingma & Ba, 2015), ADADELTA (Zeiler, 2012), NADAM (Dozat, 2016), etc, have been proposed which mitigate the rapid decay of the learning rate using the exponential moving averages of squared past gradients, essentially limiting the reliance of the update to only the past few gradients. While these algorithms have been successfully employed in several practical applications, they have also been observed to not converge in some other settings. It has been typically observed that in these settings some minibatches provide large gradients but only quite rarely, and while these large gradients are quite informative, their influence dies out rather quickly due to the exponential averaging, thus leading to poor convergence.

In this paper, we analyze this situation in detail. We rigorously prove that the intuition conveyed in the above paragraph is indeed correct; that limiting the reliance of the update on essentially only the past few gradients can indeed cause significant convergence issues. In particular, we make the following key contributions:

- We elucidate how the exponential moving average in the RMSPROP and ADAM algorithms can cause non-convergence by providing an example of simple convex optimization prob-

lem where RMSPROP and ADAM provably do not converge to an optimal solution. Our analysis easily extends to other algorithms using exponential moving averages such as ADADELTA and NADAM as well, but we omit this for the sake of clarity. In fact, the analysis is flexible enough to extend to other algorithms that employ averaging squared gradients over essentially a fixed size window (for exponential moving averages, the influences of gradients beyond a fixed window size becomes negligibly small) in the immediate past. We omit the general analysis in this paper for the sake of clarity.

- The above result indicates that in order to have guaranteed convergence the optimization algorithm must have "long-term memory" of past gradients. Specifically, we point out a problem with the proof of convergence of the ADAM algorithm given by Kingma & Ba (2015). To resolve this issue, we propose new variants of ADAM which rely on long-term memory of past gradients, but can be implemented in the same time and space requirements as the original ADAM algorithm. We provide a convergence analysis for the new variants in the convex setting, based on the analysis of Kingma & Ba (2015), and show a data-dependent regret bound similar to the one in ADAGRAD.

- We provide a preliminary empirical study of one of the variants we proposed and show that it either performs similarly, or better, on some commonly used problems in machine learning.

## 2 PRELIMINARIES

**Notation.** We use $\mathcal{S}_d^+$ to denote the set of all positive definite $d \times d$ matrices. With slight abuse of notation, for a vector $a \in \mathbb{R}^d$ and a positive definite matrix $M \in \mathbb{R}^d \times \mathbb{R}^d$, we use $a/M$ to denote $M^{-1}a$, $\|M_i\|_2$ to denote $\ell_2$-norm of $i^{th}$ row of M and $\sqrt{M}$ to represent $M^{1/2}$. Furthermore, for any vectors $a, b \in \mathbb{R}^d$, we use $\sqrt{a}$ for element-wise square root, $a^2$ for element-wise square, $a/b$ to denote element-wise division and $\max(a, b)$ to denote element-wise maximum. For any vector $\theta_i \in \mathbb{R}^d$, $\theta_{i,j}$ denotes its $j^{\text{th}}$ coordinate where $j \in [d]$. The projection operation $\Pi_{\mathcal{F},A}(y)$ for $A \in \mathcal{S}_+^d$ is defined as $\arg\min_{x \in \mathcal{F}} \|A^{1/2}(x - y)\|$ for $y \in \mathbb{R}^d$. Finally, we say $\mathcal{F}$ has bounded diameter $D_\infty$ if $\|x - y\|_\infty \leq D_\infty$ for all $x, y \in \mathcal{F}$.

**Optimization setup.** A flexible framework to analyze iterative optimization methods is the online optimization problem in the full information feedback setting. In this online setup, at each time step $t$, the optimization algorithm picks a point (i.e. the parameters of the model to be learned) $x_t \in \mathcal{F}$, where $\mathcal{F} \in \mathbb{R}^d$ is the feasible set of points. A loss function $f_t$ (to be interpreted as the loss of the model with the chosen parameters in the next minibatch) is then revealed, and the algorithm incurs loss $f_t(x_t)$. The algorithm's regret at the end of $T$ rounds of this process is given by $R_T = \sum_{i=1}^T f_t(x_t) - \min_{x \in \mathcal{F}} \sum_{i=1}^T f_t(x)$. Throughout this paper, we assume that the feasible set $\mathcal{F}$ has bounded diameter and $\|\nabla f_t(x)\|_\infty$ is bounded for all $t \in [T]$ and $x \in \mathcal{F}$.

Our aim to is to devise an algorithm that ensures $R_T = o(T)$, which implies that on average, the model's performance converges to the optimal one. The simplest algorithm for this setting is the standard online gradient descent algorithm (Zinkevich, 2003), which moves the point $x_t$ in the opposite direction of the gradient $g_t = \nabla f_t(x_t)$ while maintaining the feasibility by projecting onto the set $\mathcal{F}$ via the update rule $x_{t+1} = \Pi_{\mathcal{F}}(x_t - \alpha_t g_t)$, where $\Pi_{\mathcal{F}}(y)$ denotes the projection of $y \in \mathbb{R}^d$ onto the set $\mathcal{F}$ i.e., $\Pi_{\mathcal{F}}(y) = \min_{x \in \mathcal{F}} \|x - y\|$, and $\alpha_t$ is typically set to $\alpha/\sqrt{t}$ for some constant $\alpha$. The aforementioned online learning problem is closely related to the stochastic optimization problem: $\min_{x \in \mathcal{F}} \mathbb{E}_z[f(x, z)]$, popularly referred to as empirical risk minimization (ERM), where $z$ is a training example drawn training sample over which a model with parameters $x$ is to be learned, and $f(x, z)$ is the loss of the model with parameters $x$ on the sample $z$. In particular, an online optimization algorithm with vanishing average regret yields a stochastic optimization algorithm for the ERM problem (Cesa-Bianchi et al., 2004). Thus, we use online gradient descent and stochastic gradient descent (SGD) synonymously.

**Generic adaptive methods setup.** We now provide a framework of adaptive methods that gives us insights into the differences between different adaptive methods and is useful for understanding the flaws in a few popular adaptive methods. Algorithm 1 provides a generic adaptive framework that encapsulates many popular adaptive methods. Note the algorithm is still abstract because the

---

**Algorithm 1** Generic Adaptive Method Setup

---

**Input:** $x_1 \in \mathcal{F}$, step size $\{\alpha_t > 0\}_{t=1}^T$, sequence of functions $\{\phi_t, \psi_t\}_{t=1}^T$
**for** $t = 1$ **to** $T$ **do**
    $g_t = \nabla f_t(x_t)$
    $m_t = \phi_t(g_1, \ldots, g_t)$ and $V_t = \psi_t(g_1, \ldots, g_t)$
    $\hat{x}_{t+1} = x_t - \alpha_t m_t / \sqrt{V_t}$
    $x_{t+1} = \Pi_{\mathcal{F}, \sqrt{V}_t}(\hat{x}_{t+1})$
**end for**

---

"averaging" functions $\phi_t$ and $\psi_t$ have not been specified. Here $\phi_t : \mathcal{F}^t \to \mathbb{R}^d$ and $\psi_t : \mathcal{F}^t \to \mathcal{S}_+^d$. For ease of exposition, we refer to $\alpha_t$ as step size and $\alpha_t V_t^{-1/2}$ as learning rate of the algorithm and furthermore, restrict ourselves to diagonal variants of adaptive methods encapsulated by Algorithm 1 where $V_t = \text{diag}(v_t)$ . We first observe that standard stochastic gradient algorithm falls in this framework by using:

$$\phi_t(g_1, \ldots, g_t) = g_t \text{ and } \psi_t(g_1, \ldots, g_t) = \mathbb{I}, \tag{SGD}$$

and $\alpha_t = \alpha / \sqrt{t}$ for all $t \in [T]$. While the decreasing step size is required for convergence, such an aggressive decay of learning rate typically translates into poor empirical performance. The key idea of adaptive methods is to choose averaging functions appropriately so as to entail good convergence. For instance, the first adaptive method ADAGRAD (Duchi et al., 2011), which propelled the research on adaptive methods, uses the following averaging functions:

$$\phi_t(g_1, \ldots, g_t) = g_t \text{ and } \psi_t(g_1, \ldots, g_t) = \frac{\text{diag}(\sum_{i=1}^t g_i^2)}{t}, \tag{ADAGRAD}$$

and step size $\alpha_t = \alpha / \sqrt{t}$ for all $t \in [T]$. In contrast to a learning rate of $\alpha / \sqrt{t}$ in SGD, such a setting effectively implies a modest learning rate decay of $\alpha / \sqrt{\sum_i g_{i,j}^2}$ for $j \in [d]$. When the gradients are sparse, this can potentially lead to huge gains in terms of convergence (see Duchi et al. (2011)). These gains have also been observed in practice for even few non-sparse settings.

**Adaptive methods based on Exponential Moving Averages.** Exponential moving average variants of ADAGRAD are popular in the deep learning community. RMSPROP, ADAM, NADAM, and ADADELTA are some prominent algorithms that fall in this category. The key difference is to use an exponential moving average as function $\psi_t$ instead of the simple average function used in ADAGRAD. ADAM[1], a particularly popular variant, uses the following averaging functions:

$$\phi_t(g_1, \ldots, g_t) = (1 - \beta_1) \sum_{i=1}^t \beta_1^{t-i} g_i \text{ and } \psi_t(g_1, \ldots, g_t) = (1 - \beta_2) \text{diag}(\sum_{i=1}^t \beta_2^{t-i} g_i^2), \tag{ADAM}$$

for some $\beta_1, \beta_2 \in [0, 1)$. This update can alternatively be stated by the following simple recursion:

$$m_{t,i} = \beta_1 m_{t-1,i} + (1 - \beta_1) g_{t,i} \text{ and } v_{t,i} = \beta_2 v_{t-1,i} + (1 - \beta_2) g_{t,i}^2 \tag{1}$$

and $m_{0,i} = 0$ and $v_{0,i} = 0$ for all $i \in [d]$. and $t \in [T]$. A value of $\beta_1 = 0.9$ and $\beta_2 = 0.999$ is typically recommended in practice. We note the additional projection operation in Algorithm 1 in comparison to ADAM. When $\mathcal{F} = \mathbb{R}^d$, the projection operation is an identity operation and this corresponds to the algorithm in (Kingma & Ba, 2015). For theoretical analysis, one requires $\alpha_t = 1/\sqrt{t}$ for $t \in [T]$, although, a more aggressive choice of constant step size seems to work well in practice. RMSPROP, which appeared in an earlier unpublished work (Tieleman & Hinton, 2012) is essentially a variant of ADAM with $\beta_1 = 0$. In practice, especially in deep learning applications, the momentum term arising due to non-zero $\beta_1$ appears to significantly boost the performance. We will mainly focus on ADAM algorithm due to this generality but our arguments also apply to RMSPROP and other algorithms such as ADADELTA, NADAM.

---

[1]Here, for simplicity, we remove the debiasing step used in the version of ADAM used in the original paper by Kingma & Ba (2015). However, our arguments also apply to the debiased version as well.

## 3 THE NON-CONVERGENCE OF ADAM

With the problem setup in the previous section, we discuss fundamental flaw in the current exponential moving average methods like ADAM. We show that ADAM can fail to converge to an optimal solution even in simple one-dimensional convex settings. These examples of non-convergence contradict the claim of convergence in (Kingma & Ba, 2015), and the main issue lies in the following quantity of interest:

$$\Gamma_{t+1} = \left( \frac{\sqrt{V_{t+1}}}{\alpha_{t+1}} - \frac{\sqrt{V_t}}{\alpha_t} \right). \tag{2}$$

This quantity essentially measures the change in the inverse of learning rate of the adaptive method with respect to time. One key observation is that for SGD and ADAGRAD, $\Gamma_t \succeq 0$ for all $t \in [T]$. This simply follows from update rules of SGD and ADAGRAD in the previous section. In particular, update rules for these algorithms lead to "non-increasing" learning rates. However, this is not necessarily the case for exponential moving average variants like ADAM and RMSPROP i.e., $\Gamma_t$ can potentially be indefinite for $t \in [T]$. We show that this violation of positive definiteness can lead to undesirable convergence behavior for ADAM and RMSPROP. Consider the following simple sequence of linear functions for $\mathcal{F} = [-1, 1]$:

$$f_t(x) = \begin{cases} Cx, & \text{for } t \bmod 3 = 1 \\ -x, & \text{otherwise}, \end{cases}$$

where $C > 2$. For this function sequence, it is easy to see that the point $x = -1$ provides the minimum regret. Suppose $\beta_1 = 0$ and $\beta_2 = 1/(1 + C^2)$. We show that ADAM converges to a highly suboptimal solution of $x = +1$ for this setting. Intuitively, the reasoning is as follows. The algorithm obtains the large gradient $C$ once every 3 steps, and while the other 2 steps it observes the gradient $-1$, which moves the algorithm in the wrong direction. The large gradient $C$ is unable to counteract this effect since it is scaled down by a factor of almost $C$ for the given value of $\beta_2$, and hence the algorithm converges to 1 rather than $-1$. We formalize this intuition in the result below.

**Theorem 1.** *There is an online convex optimization problem where* ADAM *has non-zero average regret i.e., $R_T/T \nrightarrow 0$ as $T \to \infty$.*

We relegate all proofs to the appendix. A few remarks are in order. One might wonder if adding a small constant in the denominator of the update helps in circumventing this problem i.e., the update for ADAM in Algorithm 1 of $\hat{x}_{t+1}$ is modified as follows:

$$\hat{x}_{t+1} = x_t - \alpha_t m_t / \sqrt{V_t + \epsilon \mathbb{I}}. \tag{3}$$

The algorithm in (Kingma & Ba, 2015) uses such an update in practice, although their analysis does not. In practice, selection of the $\epsilon$ parameter appears to be critical for the performance of the algorithm. However, we show that for any constant $\epsilon > 0$, there exists an online optimization setting where, again, ADAM has non-zero average regret asymptotically (see Theorem 6 in Section F of the appendix).

The above examples of non-convergence are catastrophic insofar that ADAM and RMSPROP converge to a point that is worst amongst all points in the set $[-1, 1]$. Note that above example also holds for constant step size $\alpha_t = \alpha$. Also note that classic SGD and ADAGRAD do not suffer from this problem and for these algorithms, average regret asymptotically goes to 0. This problem is especially aggravated in high dimensional settings and when the variance of the gradients with respect to time is large. This example also provides intuition for why large $\beta_2$ is advisable while using ADAM algorithm, and indeed in practice using large $\beta_2$ helps. However the following result shows that for any constant $\beta_1$ and $\beta_2$ with $\beta_1 < \sqrt{\beta_2}$, we can design an example where ADAM has non-zero average rate asymptotically.

**Theorem 2.** *For any constant $\beta_1, \beta_2 \in [0, 1)$ such that $\beta_1 < \sqrt{\beta_2}$, there is an online convex optimization problem where* ADAM *has non-zero average regret i.e., $R_T/T \nrightarrow 0$ as $T \to \infty$.*

The above results show that with constant $\beta_1$ and $\beta_2$, momentum or regularization via $\epsilon$ will not help in convergence of the algorithm to the optimal solution. Note that the condition $\beta_1 < \sqrt{\beta_2}$ is benign and is typically satisfied in the parameter settings used in practice. Furthermore, such condition is assumed in convergence proof of Kingma & Ba (2015). We can strengthen this result by providing a similar example of non-convergence even in the easier stochastic optimization setting:

---

**Algorithm 2** AMSGRAD

**Input:** $x_1 \in \mathcal{F}$, step size $\{\alpha_t\}_{t=1}^T$, $\{\beta_{1t}\}_{t=1}^T$, $\beta_2$
Set $m_0 = 0$, $v_0 = 0$ and $\hat{v}_0 = 0$
**for** $t = 1$ **to** $T$ **do**
    $g_t = \nabla f_t(x_t)$
    $m_t = \beta_{1t} m_{t-1} + (1 - \beta_{1t}) g_t$
    $v_t = \beta_2 v_{t-1} + (1 - \beta_2) g_t^2$
    $\hat{v}_t = \max(\hat{v}_{t-1}, v_t)$ and $\hat{V}_t = \text{diag}(\hat{v}_t)$
    $x_{t+1} = \Pi_{\mathcal{F}, \sqrt{\hat{V}_t}}(x_t - \alpha_t m_t / \sqrt{\hat{v}_t})$
**end for**

---

**Theorem 3.** *For any constant $\beta_1, \beta_2 \in [0, 1)$ such that $\beta_1 < \sqrt{\beta_2}$, there is a stochastic convex optimization problem for which* ADAM *does not converge to the optimal solution.*

These results have important consequences insofar that one has to use "problem-dependent" $\epsilon, \beta_1$ and $\beta_2$ in order to avoid bad convergence behavior. In high-dimensional problems, this typically amounts to using, unlike the update in Equation (3), a different $\epsilon, \beta_1$ and $\beta_2$ for each dimension. However, this defeats the purpose of adaptive methods since it requires tuning a large set of parameters. We would also like to emphasize that while the example of non-convergence is carefully constructed to demonstrate the problems in ADAM, it is not unrealistic to imagine scenarios where such an issue can at the very least slow down convergence.

We end this section with the following important remark. While the results stated above use constant $\beta_1$ and $\beta_2$, the analysis of ADAM in (Kingma & Ba, 2015) actually relies on decreasing $\beta_1$ over time. It is quite easy to extend our examples to the case where $\beta_1$ is decreased over time, since the critical parameter is $\beta_2$ rather than $\beta_1$, and as long as $\beta_2$ is bounded away from 1, our analysis goes through. Thus for the sake of clarity, in this paper we only prove non-convergence of ADAM in the setting where $\beta_1$ is held constant.

## 4 A NEW EXPONENTIAL MOVING AVERAGE VARIANT: AMSGRAD

In this section, we develop a new principled exponential moving average variant and provide its convergence analysis. Our aim is to devise a new strategy with guaranteed convergence while preserving the practical benefits of ADAM and RMSPROP. To understand the design of our algorithms, let us revisit the quantity $\Gamma_t$ in (2). For ADAM and RMSPROP, this quantity can potentially be negative. The proof in the original paper of ADAM erroneously assumes that $\Gamma_t$ is positive semi-definite and is hence, incorrect (refer to Appendix D for more details). For the first part, we modify these algorithms to satisfy this additional constraint. Later on, we also explore an alternative approach where $\Gamma_t$ can be made positive semi-definite by using values of $\beta_1$ and $\beta_2$ that change with $t$.

AMSGRAD uses a smaller learning rate in comparison to ADAM and yet incorporates the intuition of slowly decaying the effect of past gradients on the learning rate as long as $\Gamma_t$ is positive semi-definite. Algorithm 2 presents the pseudocode for the algorithm. The key difference of AMSGRAD with ADAM is that it maintains the maximum of all $v_t$ until the present time step and uses this maximum value for normalizing the running average of the gradient instead of $v_t$ in ADAM. By doing this, AMSGRAD results in a non-increasing step size and avoids the pitfalls of ADAM and RMSPROP i.e., $\Gamma_t \succeq 0$ for all $t \in [T]$ even with constant $\beta_2$. Also, in Algorithm 2, one typically uses a constant $\beta_{1t}$ in practice (although, the proof requires a decreasing schedule for proving convergence of the algorithm).

To gain more intuition for the updates of AMSGRAD, it is instructive to compare its update with ADAM and ADAGRAD. Suppose at particular time step $t$ and coordinate $i \in [d]$, we have $v_{t-1,i} > g_{t,i}^2 > 0$, then ADAM aggressively increases the learning rate, however, as we have seen in the previous section, this can be detrimental to the overall performance of the algorithm. On the other hand, ADAGRAD slightly decreases the learning rate, which often leads to poor performance in practice since such an accumulation of gradients over a large time period can significantly decrease the learning rate. In contrast, AMSGRAD neither increases nor decreases the learning rate and furthermore, decreases $v_t$ which can potentially lead to non-decreasing learning rate even if gradient

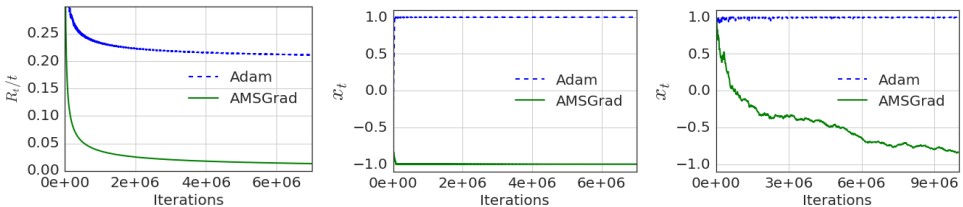

Figure 1: Performance comparison of ADAM and AMSGRAD on synthetic example on a simple one dimensional convex problem inspired by our examples of non-convergence. The first two plots (left and center) are for the online setting and the the last one (right) is for the stochastic setting.

is large in the future iterations. For rest of the paper, we use $g_{1:t} = [g_1 \ldots g_t]$ to denote the matrix obtained by concatenating the gradient sequence. We prove the following key result for AMSGRAD.

**Theorem 4.** *Let $\{x_t\}$ and $\{v_t\}$ be the sequences obtained from Algorithm 2, $\alpha_t = \alpha/\sqrt{t}$, $\beta_1 = \beta_{11}$, $\beta_{1t} \leq \beta_1$ for all $t \in [T]$ and $\gamma = \beta_1/\sqrt{\beta_2} < 1$. Assume that $\mathcal{F}$ has bounded diameter $D_\infty$ and $\|\nabla f_t(x)\|_\infty \leq G_\infty$ for all $t \in [T]$ and $x \in \mathcal{F}$. For $x_t$ generated using the AMSGRAD (Algorithm 2), we have the following bound on the regret*

$$R_T \leq \frac{D_\infty^2 \sqrt{T}}{\alpha(1-\beta_1)} \sum_{i=1}^d \hat{v}_{T,i}^{1/2} + \frac{D_\infty^2}{2(1-\beta_1)} \sum_{t=1}^T \sum_{i=1}^d \frac{\beta_{1t}\hat{v}_{t,i}^{1/2}}{\alpha_t} + \frac{\alpha\sqrt{1+\log T}}{(1-\beta_1)^2(1-\gamma)\sqrt{(1-\beta_2)}} \sum_{i=1}^d \|g_{1:T,i}\|_2.$$

The following result falls as an immediate corollary of the above result.

**Corollary 1.** *Suppose $\beta_{1t} = \beta_1\lambda^{t-1}$ in Theorem 4, then we have*

$$R_T \leq \frac{D_\infty^2 \sqrt{T}}{\alpha(1-\beta_1)} \sum_{i=1}^d \hat{v}_{T,i}^{1/2} + \frac{\beta_1 D_\infty^2 G_\infty}{2(1-\beta_1)(1-\lambda)^2} + \frac{\alpha\sqrt{1+\log T}}{(1-\beta_1)^2(1-\gamma)\sqrt{(1-\beta_2)}} \sum_{i=1}^d \|g_{1:T,i}\|_2.$$

The above bound can be considerably better than $O(\sqrt{dT})$ regret of SGD when $\sum_{i=1}^d \hat{v}_{T,i}^{1/2} \ll \sqrt{d}$ and $\sum_{i=1}^d \|g_{1:T,i}\|_2 \ll \sqrt{dT}$ (Duchi et al., 2011). Furthermore, in Theorem 4, one can use a much more modest momentum decay of $\beta_{1t} = \beta_1/t$ and still ensure a regret of $O(\sqrt{T})$. We would also like to point out that one could consider taking a simple average of all the previous values of $v_t$ instead of their maximum. The resulting algorithm is very similar to ADAGRAD except for normalization with smoothed gradients rather than actual gradients and can be shown to have similar convergence as ADAGRAD.

## 5 EXPERIMENTS

In this section, we present empirical results on both synthetic and real-world datasets. For our experiments, we study the problem of multiclass classification using logistic regression and neural networks, representing convex and nonconvex settings, respectively.

**Synthetic Experiments**: To demonstrate the convergence issue of ADAM, we first consider the following simple convex setting inspired from our examples of non-convergence:

$$f_t(x) = \begin{cases} 1010x, & \text{for } t \bmod 101 = 1 \\ -10x, & \text{otherwise,} \end{cases}$$

with the constraint set $\mathcal{F} = [-1, 1]$. We first observe that, similar to the examples of non-convergence we have considered, the optimal solution is $x = -1$; thus, for convergence, we expect the algorithms to converge to $x = -1$. For this sequence of functions, we investigate the regret and the value of the iterate $x_t$ for ADAM and AMSGRAD. To enable fair comparison, we set $\beta_1 = 0.9$ and $\beta_2 = 0.99$ for ADAM and AMSGRAD algorithm, which are typically the parameters settings used for ADAM in practice. Figure 1 shows the average regret ($R_t/t$) and value of the iterate ($x_t$) for

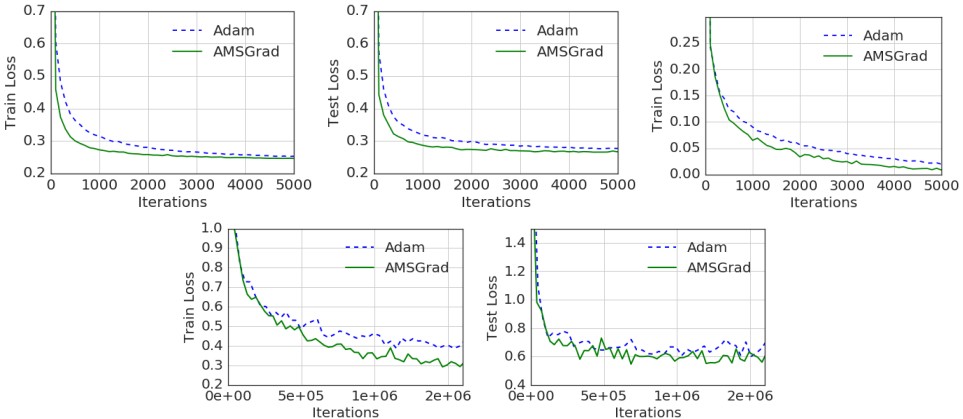

Figure 2: Performance comparison of ADAM and AMSGRAD for logistic regression, feedforward neural network and CIFARNET. The top row shows performance of ADAM and AMSGRAD on logistic regression (left and center) and 1-hidden layer feedforward neural network (right) on MNIST. In the bottom row, the two plots compare the training and test loss of ADAM and AMSGRAD with respect to the iterations for CIFARNET.

this problem. We first note that the average regret of ADAM does not converge to $0$ with increasing $t$. Furthermore, its iterates $x_t$ converge to $x = 1$, which unfortunately has the largest regret amongst all points in the domain. On the other hand, the average regret of AMSGRAD converges to $0$ and its iterate converges to the optimal solution. Figure 1 also shows the stochastic optimization setting:

$$f_t(x) = \begin{cases} 1010x, & \text{with probability } 0.01 \\ -10x, & \text{otherwise.} \end{cases}$$

Similar to the aforementioned online setting, the optimal solution for this problem is $x = -1$. Again, we see that the iterate $x_t$ of ADAM converges to the highly suboptimal solution $x = 1$.

**Logistic Regression**: To investigate the performance of the algorithm on convex problems, we compare AMSGRAD with ADAM on logistic regression problem. We use MNIST dataset for this experiment, the classification is based on $784$ dimensional image vector to one of the $10$ class labels. The step size parameter $\alpha_t$ is set to $\alpha/\sqrt{t}$ for both ADAM and AMSGRAD in for our experiments, consistent with the theory. We use a minibatch version of these algorithms with minibatch size set to $128$. We set $\beta_1 = 0.9$ and $\beta_2$ is chosen from the set $\{0.99, 0.999\}$, but they are fixed throughout the experiment. The parameters $\alpha$ and $\beta_2$ are chosen by grid search. We report the train and test loss with respect to iterations in Figure 2. We can see that AMSGRAD performs better than ADAM with respect to both train and test loss. We also observed that AMSGRAD is relatively more robust to parameter changes in comparison to ADAM.

**Neural Networks**: For our first experiment, we trained a simple 1-hidden fully connected layer neural network for the multiclass classification problem on MNIST. Similar to the previous experiment, we use $\beta_1 = 0.9$ and $\beta_2$ is chosen from $\{0.99, 0.999\}$. We use a fully connected 100 rectified linear units (ReLU) as the hidden layer for this experiment. Furthermore, we use constant $\alpha_t = \alpha$ throughout all our experiments on neural networks. Such a parameter setting choice of ADAM is consistent with the ones typically used in the deep learning community for training neural networks. A grid search is used to determine parameters that provides the best performance for the algorithm.

Finally, we consider the multiclass classification problem on the standard CIFAR-10 dataset, which consists of $60,000$ labeled examples of $32 \times 32$ images. We use CIFARNET, a convolutional neural network (CNN) with several layers of convolution, pooling and non-linear units, for training a multiclass classifer for this problem. In particular, this architecture has 2 convolutional layers with 64 channels and kernel size of $6 \times 6$ followed by 2 fully connected layers of size 384 and 192. The network uses $2 \times 2$ max pooling and layer response normalization between the convolutional layers (Krizhevsky et al., 2012). A dropout layer with keep probability of $0.5$ is applied in between the fully connected layers (Srivastava et al., 2014). The minibatch size is also set to 128 similar to previous experiments. The results for this problem are reported in Figure 2. The parameters for ADAM and AMSGRAD are selected in a way similar to the previous experiments. We can see that

AMSGRAD performs considerably better than ADAM on train loss and accuracy. Furthermore, this performance gain also translates into good performance on test loss.

## 5.1 EXTENSION: ADAMNC ALGORITHM

An alternative approach is to use an increasing schedule of $\beta_2$ in ADAM. This approach, unlike Algorithm 2 does not require changing the structure of ADAM but rather uses a non-constant $\beta_1$ and $\beta_2$. The pseudocode for the algorithm, ADAMNC, is provided in the appendix (Algorithm 3). We show that by appropriate selection of $\beta_{1t}$ and $\beta_{2t}$, we can achieve good convergence rates.

**Theorem 5.** *Let $\{x_t\}$ and $\{v_t\}$ be the sequences obtained from Algorithm 3, $\alpha_t = \alpha/\sqrt{t}$, $\beta_1 = \beta_{11}$ and $\beta_{1t} \leq \beta_1$ for all $t \in [T]$. Assume that $\mathcal{F}$ has bounded diameter $D_\infty$ and $\|\nabla f_t(x)\|_\infty \leq G_\infty$ for all $t \in [T]$ and $x \in \mathcal{F}$. Furthermore, let $\{\beta_{2t}\}$ be such that the following conditions are satisfied:*

1. $\frac{1}{\alpha_T}\sqrt{\sum_{j=1}^{t} \Pi_{k=1}^{t-j}\beta_{2(t-k+1)}(1-\beta_{2j})g_{j,i}^2} \geq \frac{1}{\zeta}\sqrt{\sum_{j=1}^{t} g_{j,i}^2}$ *for some $\zeta > 0$ and all $t \in [T]$, $j \in [d]$.*

2. $\frac{v_{t,i}^{1/2}}{\alpha_t} \geq \frac{v_{t-1,i}^{1/2}}{\alpha_{t-1}}$ *for all $t \in \{2, \cdots, T\}$ and $i \in [d]$.*

*Then for $x_t$ generated using the ADAMNC (Algorithm 3), we have the following bound on the regret*

$$R_T \leq \frac{D_\infty^2}{2\alpha(1-\beta_1)}\sum_{i=1}^{d}\sqrt{T}v_{T,i}^{1/2} + \frac{D_\infty^2}{2(1-\beta_1)}\sum_{t=1}^{T}\sum_{i=1}^{d}\frac{\beta_{1t}v_{t,i}^{1/2}}{\alpha_t} + \frac{2\zeta}{(1-\beta_1)^3}\sum_{i=1}^{d}\|g_{1:T,i}\|_2.$$

The above result assumes selection of $\{(\alpha_t, \beta_{2t})\}$ such that $\Gamma_t \succeq 0$ for all $t \in \{2, \cdots, T\}$. However, one can generalize the result to deal with the case where this constraint is violated as long as the violation is not too large or frequent. Following is an immediate consequence of the above result.

**Corollary 2.** *Suppose $\beta_{1t} = \beta_1\lambda^{t-1}$ and $\beta_{2t} = 1 - 1/t$ in Theorem 5, then we have*

$$\frac{D_\infty^2}{2\alpha(1-\beta_1)}\sum_{i=1}^{d}\|g_{1:T,i}\|_2 + \frac{\beta_1 D_\infty^2 G_\infty}{2(1-\beta_1)(1-\lambda)^2} + \frac{2\zeta}{(1-\beta_1)^3}\sum_{i=1}^{d}\|g_{1:T,i}\|_2.$$

The above corollary follows from a trivial fact that $v_{t,i} = \sum_{j=1}^{t} g_{j,i}^2/t$ for all $i \in [d]$ when $\beta_{2t} = 1 - 1/t$. This corollary is interesting insofar that such a parameter setting effectively yields a momentum based variant of ADAGRAD. Similar to ADAGRAD, the regret is data-dependent and can be considerably better than $O(\sqrt{dT})$ regret of SGD when $\sum_{i=1}^{d}\|g_{1:T,i}\|_2 \ll \sqrt{dT}$ (Duchi et al., 2011). It is easy to generalize this result for setting similar settings of $\beta_{2t}$. Similar to Corollary 1, one can use a more modest decay of $\beta_{1t} = \beta_1/t$ and still ensure a data-dependent regret of $O(\sqrt{T})$.

## 6 DISCUSSION

In this paper, we study exponential moving variants of ADAGRAD and identify an important flaw in these algorithms which can lead to undesirable convergence behavior. We demonstrate these problems through carefully constructed examples where RMSPROP and ADAM converge to highly suboptimal solutions. In general, any algorithm that relies on an essentially fixed sized window of past gradients to scale the gradient updates will suffer from this problem.

We proposed fixes to this problem by slightly modifying the algorithms, essentially endowing the algorithms with a long-term memory of past gradients. These fixes retain the good practical performance of the original algorithms, and in some cases actually show improvements.

The primary goal of this paper is to highlight the problems with popular exponential moving average variants of ADAGRAD from a theoretical perspective. RMSPROP and ADAM have been immensely successful in development of several state-of-the-art solutions for a wide range of problems. Thus, it is important to understand their behavior in a rigorous manner and be aware of potential pitfalls while using them in practice. We believe this paper is a first step in this direction and suggests good design principles for faster and better stochastic optimization.

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

APPENDIX

## A  PROOF OF THEOREM 1

*Proof.* We consider the setting where $f_t$ are linear functions and $\mathcal{F} = [-1, 1]$. In particular, we define the following function sequence:

$$f_t(x) = \begin{cases} Cx, & \text{for } t \bmod 3 = 1 \\ -x, & \text{otherwise,} \end{cases}$$

where $C \geq 2$. For this function sequence, it is easy to see that the point $x = -1$ provides the minimum regret. Without loss of generality, assume that the initial point is $x_1 = 1$. This can be assumed without any loss of generality because for any choice of initial point, we can always translate the coordinate system such that the initial point is $x_1 = 1$ in the new coordinate system and then choose the sequence of functions as above in the new coordinate system. Also, since the problem is one-dimensional, we drop indices representing coordinates from all quantities in Algorithm 1. Consider the execution of ADAM algorithm for this sequence of functions with

$$\beta_1 = 0, \beta_2 = \frac{1}{1 + C^2} \text{ and } \alpha_t = \frac{\alpha}{\sqrt{t}}$$

where $\alpha < \sqrt{1 - \beta_2}$. Note that since gradients of these functions are bounded, $\mathcal{F}$ has bounded $L_\infty$ diameter and $\beta_1^2/\sqrt{\beta_2} < 1$. Hence, the conditions on the parameters required for ADAM are satisfied (refer to (Kingma & Ba, 2015) for more details).

Our main claim is that for iterates $\{x_t\}_{t=1}^\infty$ arising from the updates of ADAM, we have $x_t > 0$ for all $t \in \mathbb{N}$ and furthermore, $x_{3t+1} = 1$ for all $t \in \mathbb{N} \cup \{0\}$. For proving this, we resort to the principle of mathematical induction. Since $x_1 = 1$, both the aforementioned conditions hold for the base case. Suppose for some $t \in \mathbb{N} \cup \{0\}$, we have $x_i > 0$ for all $i \in [3t+1]$ and $x_{3t+1} = 1$. Our aim is to prove that $x_{3t+2}$ and $x_{3t+3}$ are positive and $x_{3t+4} = 1$. We first observe that the gradients have the following form:

$$\nabla f_i(x) = \begin{cases} C, & \text{for } i \bmod 3 = 1 \\ -1, & \text{otherwise} \end{cases}$$

From $(3t+1)^{\text{th}}$ update of ADAM in Equation (1), we obtain

$$\hat{x}_{3t+2} = x_{3t+1} - \frac{\alpha C}{\sqrt{(3t+1)(\beta_2 v_{3t} + (1 - \beta_2)C^2)}} = 1 - \frac{\alpha C}{\sqrt{(3t+1)(\beta_2 v_{3t} + (1 - \beta_2)C^2)}}.$$

The equality follows from the induction hypothesis. We observe the following:

$$\frac{\alpha C}{\sqrt{(3t+1)(\beta_2 v_{3t} + (1 - \beta_2)C^2)}} \leq \frac{\alpha C}{\sqrt{(3t+1)(1 - \beta_2)C^2}}$$
$$= \frac{\alpha}{\sqrt{(3t+1)(1 - \beta_2)}} < 1. \quad (4)$$

The second inequality follows from the step size choice that $\alpha < \sqrt{1 - \beta_2}$. Therefore, we have $0 < \hat{x}_{3t+2} < 1$ and hence $x_{3t+2} = \hat{x}_{3t+2} > 0$. Furthermore, after the $(3t+2)^{\text{th}}$ and $(3t+3)^{\text{th}}$ updates of ADAM in Equation (1), we have the following:

$$\hat{x}_{3t+3} = x_{3t+2} + \frac{\alpha}{\sqrt{(3t+2)(\beta_2 v_{3t+1} + (1 - \beta_2))}},$$
$$\hat{x}_{3t+4} = x_{3t+3} + \frac{\alpha}{\sqrt{(3t+3)(\beta_2 v_{3t+2} + (1 - \beta_2))}}.$$

Since $x_{3t+2} > 0$, it is easy to see that $x_{3t+3} > 0$. To complete the proof, we need to show that $x_{3t+4} = 1$. In order to prove this claim, we show that $\hat{x}_{3t+4} \geq 1$, which readily translates to $x_{3t+4} = 1$ because $x_{3t+4} = \Pi_\mathcal{F}(\hat{x}_{3t+4})$ and $\mathcal{F} = [-1, 1]$ here $\Pi_\mathcal{F}$ is the simple Euclidean projection (note that in one-dimension, $\Pi_{\mathcal{F}, \sqrt{V_t}} = \Pi_\mathcal{F}$). We observe the following:

$$\hat{x}_{3t+4} = \min(\hat{x}_{3t+3}, 1) + \frac{\alpha}{\sqrt{(3t+3)(\beta_2 v_{3t+2} + (1 - \beta_2))}}.$$

The above equality is due to the fact that $\hat{x}_{3t+3} > 0$ and property of projection operation onto the set $\mathcal{F} = [-1, 1]$. We consider the following two cases:

1. Suppose $\hat{x}_{3t+3} \geq 1$, then it is easy to see from the above equality that $\hat{x}_{3t+4} > 1$.

2. Suppose $\hat{x}_{3t+3} < 1$, then we have the following:

$$
\begin{aligned}
\hat{x}_{3t+4} &= \hat{x}_{3t+3} + \frac{\alpha}{\sqrt{(3t+3)(\beta_2 v_{3t+2} + (1-\beta_2))}} \\
&= x_{3t+2} + \frac{\alpha}{\sqrt{(3t+2)(\beta_2 v_{3t+1} + (1-\beta_2))}} + \frac{\alpha}{\sqrt{(3t+3)(\beta_2 v_{3t+2} + (1-\beta_2))}} \\
&= 1 - \frac{\alpha C}{\sqrt{(3t+1)(\beta_2 v_{3t} + (1-\beta_2)C^2)}} + \frac{\alpha}{\sqrt{(3t+2)(\beta_2 v_{3t+1} + (1-\beta_2))}} \\
&\qquad + \frac{\alpha}{\sqrt{(3t+3)(\beta_2 v_{3t+2} + (1-\beta_2))}}.
\end{aligned}
$$

The third equality is due to the fact that $x_{3t+2} = \hat{x}_{3t+2}$. Thus, to prove $\hat{x}_{3t+4} > 1$, it is enough to the prove:

$$
\underbrace{\frac{\alpha C}{\sqrt{(3t+1)(\beta_2 v_{3t} + (1-\beta_2)C^2)}}}_{T_1} \leq \frac{\alpha}{\sqrt{(3t+2)(\beta_2 v_{3t+1} + (1-\beta_2))}}
$$

$$
\underbrace{+ \frac{\alpha}{\sqrt{(3t+3)(\beta_2 v_{3t+2} + (1-\beta_2))}}}_{T_2}
$$

We have the following bound on term $T_1$ from Equation (4):

$$
T_1 \leq \frac{\alpha}{\sqrt{(3t+1)(1-\beta_2)}}. \tag{5}
$$

Furthermore, we lower bound $T_2$ in the following manner:

$$
\begin{aligned}
T_2 &= \frac{\alpha}{\sqrt{(3t+2)(\beta_2 v_{3t+1} + (1-\beta_2))}} + \frac{\alpha}{\sqrt{(3t+3)(\beta_2 v_{3t+2} + (1-\beta_2))}} \\
&\geq \frac{\alpha}{\sqrt{\beta_2 C^2 + (1-\beta_2)}} \left( \frac{1}{\sqrt{3t+2}} + \frac{1}{\sqrt{3t+3}} \right) \\
&\geq \frac{\alpha}{\sqrt{\beta_2 C^2 + (1-\beta_2)}} \left( \frac{1}{\sqrt{2(3t+1)}} + \frac{1}{\sqrt{2(3t+1)}} \right) \\
&= \frac{\sqrt{2}\alpha}{\sqrt{(3t+1)(\beta_2 C^2 + (1-\beta_2))}} = \frac{\alpha}{\sqrt{(3t+1)(1-\beta_2)}} \geq T_1. \tag{6}
\end{aligned}
$$

The first inequality is due to the fact that $v_t \leq C^2$ for all $t \in \mathbb{N}$. The last inequality follows from inequality in Equation (5). The last equality is due to following fact:

$$
\sqrt{\frac{\beta_2 C^2 + (1-\beta_2)}{2}} = \sqrt{1-\beta_2}
$$

for the choice of $\beta_2 = 1/(1+C^2)$. Therefore, we have $T_2 \geq T_1$ and hence, $\hat{x}_{3t+4} \geq 1$.

Therefore, from both the cases, we see that $x_{3t+4} = 1$. Therefore, by the principle of mathematical induction it holds for all $t \in \mathbb{N} \cup \{0\}$. Thus, we have

$$
f_{3t+1}(x_{3t+1}) + f_{3t+2}(x_{3t+2}) + f_{3t+2}(x_{3t+2}) - f_{3t+1}(-1) - f_{3t+2}(-1) - f_{3t+3}(-1) \geq 2C - 4 = 2C - 4.
$$

Therefore, for every 3 steps, ADAM suffers a regret of at least $2C - 4$. More specifically, $R_T \geq (2C-4)T/3$. Since $C \geq 2$, this regret can be very large and furthermore, $R_T/T \nrightarrow 0$ as $T \to \infty$, which completes the proof. $\qquad\square$

## B  PROOF OF THEOREM 2

*Proof.* The proof generalizes the optimization setting used in Theorem 1. Throughout the proof, we assume $\beta_1 < \sqrt{\beta_2}$, which is also a condition (Kingma & Ba, 2015) assume in their paper. In this proof, we consider the setting where $f_t$ are linear functions and $\mathcal{F} = [-1, 1]$. In particular, we define the following function sequence:

$$f_t(x) = \begin{cases} Cx, & \text{for } t \bmod C = 1 \\ -x, & \text{otherwise,} \end{cases}$$

where $C \in \mathbb{N}$, $C \bmod 2 = 0$ satisfies the following:

$$(1 - \beta_1)\beta_1^{C-1}C \leq 1 - \beta_1^{C-1},$$
$$\beta_2^{(C-2)/2}C^2 \leq 1,$$
$$\frac{3(1 - \beta_1)}{2\sqrt{1 - \beta_2}}\left(1 + \frac{\gamma(1 - \gamma^{C-1})}{1 - \gamma}\right) + \frac{\beta_1^{C/2-1}}{1 - \beta_1} < \frac{C}{3}, \tag{7}$$

where $\gamma = \beta_1/\sqrt{\beta_2} < 1$. It is not hard to see that these conditions hold for large constant $C$ that depends on $\beta_1$ and $\beta_2$. Since the problem is one-dimensional, we drop indices representing coordinates from all quantities in Algorithm 1. For this function sequence, it is easy to see that the point $x = -1$ provides the minimum regret since $C \geq 2$. Furthermore, the gradients have the following form:

$$\nabla f_i(x) = \begin{cases} C, & \text{for } t \bmod C = 1 \\ -1, & \text{otherwise} \end{cases}$$

Our first observation is that $m_{kC} \leq 0$ for all $k \in \mathbb{N} \cup \{0\}$. For $k = 0$, this holds trivially due to our initialization. For the general case, observe the following:

$$m_{kC+C} = -(1 - \beta_1) - (1 - \beta_1)\beta_1 - \cdots - (1 - \beta_1)\beta_1^{C-2} + (1 - \beta_1)\beta_1^{C-1}C + \beta_1^C m_{kC} \tag{8}$$
$$= -(1 - \beta_1^{C-1}) + (1 - \beta_1)\beta_1^{C-1}C + \beta_1^C m_{kC}. \tag{9}$$

If $m_{kC} \leq 0$, it can be easily shown that $m_{kC+C} \leq 0$ for our selection of $C$ in Equation (7) by using the principle of mathematical induction. With this observation we continue to the main part of the proof. Let $T'$ be such that $t + C \leq \tau^2 t$ for all $t \geq T'$ where $\tau \leq 3/2$. All our analysis focuses on iterations $t \geq T'$. Note that any regret before $T'$ is just a constant because $T'$ is independent of $T$ and thus, the average regret is negligible as $T \to \infty$. Consider an iterate at time step $t$ of the form $kC$ after $T'$. Our claim is that

$$x_{t+C} \geq \min\{x_t + c_t, 1\} \tag{10}$$

for some $c_t > 0$. To see this, consider the updates of ADAM for the particular sequence of functions we considered are:

$$x_{t+1} = \Pi_{\mathcal{F}}\left(x_t - \frac{\alpha}{\sqrt{t}}\frac{(1 - \beta_1)C + \beta_1 m_t}{\sqrt{(1 - \beta_2)C^2 + \beta_2 v_t}}\right),$$
$$x_{t+i} = \Pi_{\mathcal{F}}\left(x_{t+i-1} - \frac{\alpha}{\sqrt{t+i-1}}\frac{-(1 - \beta_1) + \beta_1 m_{t+i-1}}{\sqrt{(1 - \beta_2) + \beta_2 v_{t+i-1}}}\right) \text{ for } i \in \{2, \cdots, C\}.$$

For $i \in \{2, \cdots, C\}$, we use the following notation:

$$\delta_t = -\frac{\alpha}{\sqrt{t}}\frac{(1 - \beta_1)C + \beta_1 m_t}{\sqrt{(1 - \beta_2)C^2 + \beta_2 v_t}},$$
$$\delta_{t+i} = -\frac{\alpha}{\sqrt{t+i}}\frac{-(1 - \beta_1) + \beta_1 m_{t+i}}{\sqrt{(1 - \beta_2) + \beta_2 v_{t+i}}} \text{ for } i \in \{1, \cdots, C - 1\}.$$

Note that if $\delta_{t+j} \geq 0$ for some $j \in \{1, \cdots, C - 1\}$ then $\delta_{t+l} \geq 0$ for all $l \in \{j, \cdots, C - 1\}$. This follows from the fact that the gradient is negative for all time steps $i \in \{2, \cdots, C\}$. Using Lemma 6 for $\{x_{t+1}, \cdots, x_{t+C}\}$ and $\{\delta_t, \cdots, \delta_{t+C-1}\}$, we have the following:

$$x_{t+C} \geq \min\left\{1, x_t + \sum_{i=t}^{t+C-1} \delta_i\right\}.$$

Let $i' = C/2$. In order to prove our claim in Equation (10), we need to prove the following:

$$\delta = \sum_{i=t}^{t+C-1} \delta_i > 0.$$

To this end, we observe the following:

$$
\begin{aligned}
\sum_{i=t+1}^{t+C-1} \delta_i &= \sum_{i=1}^{C-1} -\frac{\alpha}{\sqrt{t+i}} \frac{-(1-\beta_1) + \beta_1 m_{t+i}}{\sqrt{(1-\beta_2) + \beta_2 v_{t+i}}} \\
&= \sum_{i=2}^{C} -\frac{\alpha}{\sqrt{t+i-1}} \frac{-(1-\beta_1) + (1-\beta_1)\left[\sum_{j=1}^{i-2} \beta_1^j (-1)\right] + (1-\beta_1)\beta_1^{i-1} C + \beta_1^i m_t}{\sqrt{(1-\beta_2) + \beta_2 v_{t+i-1}}} \\
&\geq \sum_{i=2}^{C} \frac{\alpha}{\sqrt{t+i-1}} \frac{(1-\beta_1) + (1-\beta_1)\left[\sum_{j=1}^{i-2} \beta_1^j\right] - (1-\beta_1)\beta_1^{i-1} C}{\sqrt{(1-\beta_2) + \beta_2 v_{t+i-1}}} \\
&\geq \sum_{i=2}^{C} \frac{\alpha}{\tau\sqrt{t}} \frac{(1-\beta_1) + (1-\beta_1)\left[\sum_{j=1}^{i-2} \beta_1^j\right]}{\sqrt{(1-\beta_2) + \beta_2 v_{t+i-1}}} - \sum_{i=2}^{C} \frac{\alpha}{\sqrt{t}} \frac{(1-\beta_1)\beta_1^{i-1} C}{\sqrt{(1-\beta_2) + \beta_2 v_{t+i-1}}} \\
&\geq \sum_{i=2}^{C} \frac{\alpha}{\tau\sqrt{t}} \frac{(1-\beta_1) + (1-\beta_1)\left[\sum_{j=1}^{i-2} \beta_1^j\right]}{\sqrt{(1-\beta_2) + \beta_2 v_{t+i-1}}} - \sum_{i=2}^{C} \frac{\alpha}{\sqrt{t}} \frac{(1-\beta_1)\beta_1^{i-1} C}{\sqrt{(1-\beta_2) + \beta_2^{i-1}(1-\beta_2)C^2}} \\
&\geq \frac{\alpha}{\tau\sqrt{t}} \sum_{i=i'}^{C} \frac{1 - \beta_1^{i-1}}{\sqrt{(1-\beta_2) + 2\beta_2}} - \frac{\alpha}{\sqrt{t}} \frac{\gamma(1-\beta_1)(1-\gamma^{C-1})}{(1-\gamma)\sqrt{(1-\beta_2)}} \\
&\geq \frac{\alpha}{\tau\sqrt{t}\sqrt{1+\beta_2}} \left(C - i' - \frac{\beta_1^{i'-1}}{1-\beta_1}\right) - \frac{\alpha}{\sqrt{t}} \frac{\gamma(1-\beta_1)(1-\gamma^{C-1})}{(1-\gamma)\sqrt{(1-\beta_2)}} \geq 0.
\end{aligned}
$$

The first equality follows from the definition of $m_{t+i+1}$. The first inequality follows from the fact that $m_t \leq 0$ when $t \mod C = 0$ (see Equation (9) and arguments based on it). The second inequality follows from the definition of $\tau$ that $t + C \leq \tau^2 t$ for all $t \geq T'$. The third inequality is due to the fact that $v_{t+i-1} \geq (1-\beta_2)\beta_2^{i-2} C^2$. The last inequality follows from our choice of $C$. The fourth inequality is due to the following upper bound that applies for all $i' \leq i \leq C$:

$$
\begin{aligned}
v_{t+i-1} &= (1-\beta_2) \sum_{j=1}^{t+i-1} \beta_2^{t+i-1-j} g_j^2 \\
&\leq (1-\beta_2) \left[\sum_{h=1}^{k} \beta_2^{t+i-1-hC} C^2 + \sum_{j=1}^{t+i-1} \beta_2^{t+i-1-j}\right] \\
&\leq (1-\beta_2) \left[\beta_2^{i'-1} C^2 \sum_{h=0}^{k-1} \beta_2^{hC} + \frac{1}{1-\beta_2}\right] \\
&\leq (1-\beta_2) \left[\frac{\beta_2^{i'-1} C^2}{1-\beta_2^C} + \frac{1}{1-\beta_2}\right] \leq 2.
\end{aligned}
$$

The first inequality follows from online problem setting for the counter-example i.e., gradient is $C$ once every $C$ iterations and $-1$ for the rest. The last inequality follows from the fact that $\beta_2^{i'-1} C^2 \leq$

1 and $\beta_2^C \leq \beta_2$. Furthermore, from the above inequality, we have

$$
\begin{aligned}
\sum_{i=t}^{t+C-1} \delta_i &\geq \delta_t + \frac{\alpha}{\tau \sqrt{t}\sqrt{1+\beta_2}} \left( C - i' - \frac{\beta_1^{i'-1}}{1-\beta_1} \right) - \frac{\alpha}{\sqrt{t}} \frac{\gamma(1-\beta_1)(1-\gamma^{C-1})}{(1-\gamma)\sqrt{(1-\beta_2)}} \\
&= -\frac{\alpha}{\sqrt{t}} \frac{(1-\beta_1)C + \beta_1 m_t}{\sqrt{(1-\beta_2)C^2 + \beta_2 v_t}} + \frac{\alpha}{\tau \sqrt{t}\sqrt{1+\beta_2}} \left( C - i' - \frac{\beta_1^{i'-1}}{1-\beta_1} \right) \\
&\qquad\qquad\qquad\qquad\qquad\qquad\qquad - \frac{\alpha}{\sqrt{t}} \frac{\gamma(1-\beta_1)(1-\gamma^{C-1})}{(1-\gamma)\sqrt{(1-\beta_2)}} \\
&\geq -\frac{\alpha}{\sqrt{t}} \frac{(1-\beta_1)C}{\sqrt{(1-\beta_2)C^2}} + \frac{\alpha}{\tau \sqrt{t}\sqrt{1+\beta_2}} \left( C - i' - \frac{\beta_1^{i'-1}}{1-\beta_1} \right) - \frac{\alpha}{\sqrt{t}} \frac{\gamma(1-\beta_1)(1-\gamma^{C-1})}{(1-\gamma)\sqrt{(1-\beta_2)}} \\
&\geq \frac{\alpha}{\tau \sqrt{t}} \left[ \frac{C}{3} - \frac{\beta_1^{C/2-1}}{1-\beta_1} - \frac{3(1-\beta_1)}{2\sqrt{1-\beta_2}} \left( 1 + \frac{\gamma(1-\gamma^{C-1})}{1-\gamma} \right) \right] = \frac{\alpha}{\sqrt{t}} \lambda
\end{aligned}
$$

Note that from our choice of $C$, it is easy to see that $\lambda \geq 0$. Also, observe that $\lambda$ is independent of $t$. Thus, $x_{t+C} \geq \min\{1, x_t + \lambda/\sqrt{t}\}$. From this fact, we also see the following:

1. If $x_t = 1$, then $x_{t+C} = 1$ for all $t \geq T'$ such that $t \mod C = 0$.

2. There exists constant $T_1' \geq T'$ such that $x_{T_1'} = 1$ where $T_1' \mod C = 0$.

The first point simply follows from the relation $x_{t+C} \geq \min\{1, x_t + \lambda/\sqrt{t}\}$. The second point is due to divergent nature of the sum $\sum_{t=t'}^{\infty} 1/\sqrt{t}$. Therefore, we have

$$
\sum_{i=1}^{C} f_{(kC+i)}(x_{kC+i}) - \sum_{i=1}^{C} f_{(kC+i)}(-1) \geq 2C - 2(C-1) = 2.
$$

where $kC \geq T_1'$. Thus, when $t \geq T_1'$, for every C steps, ADAM suffers a regret of at least 2. More specifically, $R_T \geq 2(T - T_1')/C$. Thus, $R_T/T \nrightarrow 0$ as $T \to \infty$, which completes the proof. $\qquad\square$

## C  PROOF OF THEOREM 3

*Proof.* Let $\delta$ be an arbitrary small positive constant, and $C$ be a large enough constant chosen as a function of $\beta_1, \beta_2, \delta$ that will be determined in the proof.

Consider the following one dimensional stochastic optimization setting over the domain $[-1, 1]$. At each time step $t$, the function $f_t(x)$ is chosen i.i.d. as follows:

$$
f_t(x) = \begin{cases} Cx & \text{with probability } p := \frac{1+\delta}{C+1} \\ -x & \text{with probability } 1-p \end{cases}
$$

The expected function is $F(x) = \delta x$; thus the optimum point over $[-1, 1]$ is $x^\star = -1$. At each time step $t$ the gradient $g_t$ equals $C$ with probability $p$ and $-1$ with probability $1-p$. Thus, the step taken by ADAM is

$$
\Delta_t = \frac{-\alpha_t(\beta_1 m_{t-1} + (1-\beta_1)g_t)}{\sqrt{\beta_2 v_{t-1} + (1-\beta_2)g_t^2}}.
$$

We now show that for a large enough constant $C$, $\mathbb{E}[\Delta_t] \geq 0$, which implies that the ADAM's steps keep drifting away from the optimal solution $x^\star = -1$.

**Lemma 1.** *For a large enough constant $C$ (as a function of $\beta_1, \beta_2, \delta$), we have $\mathbb{E}[\Delta_t] \geq 0$.*

*Proof.* Let $\mathbb{E}_t[\cdot]$ denote expectation conditioned on all randomness up to and including time $t - 1$. Taking conditional expectation of the step, we have

$$\frac{1}{\alpha_t}\mathbb{E}_t[\Delta_t] = p \cdot \frac{-(\beta_1 m_{t-1} + (1-\beta_1)C)}{\sqrt{\beta_2 v_{t-1} + (1-\beta_2)C^2}} + (1-p) \cdot \frac{-(\beta_1 m_{t-1} - (1-\beta_1))}{\sqrt{\beta_2 v_{t-1} + (1-\beta_2)}}$$

$$= p \cdot \underbrace{\frac{-(\beta_1 m_{t-1} + (1-\beta_1)C)}{\sqrt{\beta_2 v_{t-1} + (1-\beta_2)C^2}}}_{T_1} + (1-p) \cdot \underbrace{\frac{-\beta_1 m_{t-1}}{\sqrt{\beta_2 v_{t-1} + (1-\beta_2)}}}_{T_2} + (1-p) \cdot \underbrace{\frac{1-\beta_1}{\sqrt{\beta_2 v_{t-1} + (1-\beta_2)}}}_{T_3}$$

$$(11)$$

We will bound the expectation of the terms $T_1$, $T_2$ and $T_3$ above separately.

First, for $T_1$, we have

$$T_1 \geq \frac{-(\beta_1 C + (1-\beta_1)C)}{\sqrt{(1-\beta_2)C^2}} \geq -\frac{1}{\sqrt{1-\beta_2}}. \tag{12}$$

Next, we bound $\mathbb{E}[T_2]$. Define $k = \lceil \frac{\log(C+1)}{\log(1/\beta_1)} \rceil$. This choice of $k$ ensures that $\beta_1^k C \leq 1 - \beta_1^k$. Now, note that

$$m_{t-1} = (1-\beta_1)\sum_{i=1}^{t-1}\beta_1^{t-1-i}g_i.$$

Let $E$ denote the event that for every $i = t-1, t-2, \ldots, \max\{t-k, 1\}$, $g_i = -1$. Note that $\Pr[E] \geq 1 - kp$. Assuming $E$ happens, we can bound $m_{t-1}$ as follows:

$$m_{t-1} \leq (1-\beta_1)\sum_{i=\max\{t-k,1\}}^{t-1}\beta_1^{t-1-i}\cdot -1 + (1-\beta_1)\sum_{i=1}^{\max\{t-k,1\}-1}\beta_1^{t-1-i}\cdot C \leq -(1-\beta_1^k) + \beta_1^k C \leq 0,$$

and so $T_2 \geq 0$.

With probability at most $kp$, the event $E$ doesn't happen. In this case, we bound $T_2$ as follows. We first bound $m_{t-1}$ in terms of $v_{t-1}$ using the Cauchy-Schwarz inequality as follows:

$$m_{t-1} = (1-\beta_1)\sum_{i=1}^{t-1}\beta_1^{t-1-i}g_i \leq (1-\beta_1)\sqrt{\left(\sum_{i=1}^{t-1}\beta_2^{t-1-i}g_i^2\right)\left(\sum_{i=1}^{t-1}(\frac{\beta_1^2}{\beta_2})^{t-1-i}\right)}$$

$$\leq \underbrace{(1-\beta_1)\sqrt{\frac{\beta_2}{(1-\beta_2)(\beta_2-\beta_1^2)}}}_{A}\cdot\sqrt{v_{t-1}}.$$

Thus, $v_{t-1} \geq m_{t-1}^2/A^2$. Thus, we have

$$T_2 = \frac{-\beta_1 m_{t-1}}{\sqrt{\beta_2 v_{t-1} + (1-\beta_2)}} \geq \frac{-\beta_1 |m_{t-1}|}{\sqrt{\beta_2(m_{t-1}^2/A^2)}} = \frac{-\beta_1(1-\beta_1)}{\sqrt{(1-\beta_2)(\beta_2-\beta_1^2)}}.$$

Hence, we have

$$\mathbb{E}[T_2] \geq 0 \cdot (1-kp) + \frac{-\beta_1(1-\beta_1)}{\sqrt{(1-\beta_2)(\beta_2-\beta_1^2)}}\cdot kp = \frac{-\beta_1(1-\beta_1)kp}{\sqrt{(1-\beta_2)(\beta_2-\beta_1^2)}} \tag{13}$$

Finally, we lower bound $\mathbb{E}[T_3]$ using Jensen's inequality applied to the convex function $\frac{1}{\sqrt{x}}$:

$$\mathbb{E}[T_3] \geq \frac{(1-\beta_1)}{\sqrt{\beta_2\mathbb{E}[v_{t-1}] + (1-\beta_2)}} \geq \frac{(1-\beta_1)}{\sqrt{\beta_2(1+\delta)C^2 + (1-\beta_2)}}.$$

The last inequality follows by using the facts $v_{t-1} = (1-\beta_2)\sum_{i=1}^{t-1}\beta_2^{t-1-i}g_i^2$, and the random variables $g_1^2, g_2^2, \ldots, g_{t-1}^2$ are i.i.d., and so

$$\mathbb{E}[v_{t-1}] = (1-\beta_2^{t-1})\mathbb{E}[g_1^2] = (1-\beta_2^{t-1})(C^2 p + (1-p)) = (1-\beta_2^{t-1})(1+\delta)C - \delta \leq (1+\delta)C. \tag{14}$$

Combining the bounds in (12), (13), and (14) in the expression for ADAM's step, (11), and plugging in the values of the parameters $k$ and $p$ we get the following lower bound on $\mathbb{E}[\Delta_t]$:

$$-\frac{1+\delta}{C+1} \cdot \left( \frac{1}{\sqrt{1-\beta_2}} + \frac{-\beta_1(1-\beta_1)\lceil\frac{\log(C+1)}{\log(1/\beta_1)}\rceil}{\sqrt{(1-\beta_2)(\beta_2-\beta_1^2)}} \right) + \left(1 - \frac{1+\delta}{C+1}\right) \cdot \frac{(1-\beta_1)}{\sqrt{\beta_2(1+\delta)C + (1-\beta_2)}}.$$

It is evident that for $C$ large enough (as a function of $\delta, \beta_1, \beta_2$), the above expression can be made non-negative. ☐

For the sake of simplicity, let us assume, as is routinely done in practice, that we are using a version of ADAM that doesn't perform any projection steps[2]. Then the lemma implies that $\mathbb{E}[x_{t+1}] \geq \mathbb{E}[x_t]$. Via a simple induction, we conclude that $\mathbb{E}[x_t] \geq x_1$ for all $t$. Thus, if we assume that the starting point $x_1 \geq 0$, then $\mathbb{E}[x_t] \geq 0$. Since $F$ is a monotonically increasing function, we have $\mathbb{E}[F(x_t)] \geq F(0) = 0$, whereas $F(-1) = -\delta$. Thus the expected suboptimality gap is always $\delta > 0$, which implies that ADAM doesn't converge to the optimal solution. ☐

## D    PROOF OF THEOREM 4

The proof of Theorem 4 presented below is along the lines of the Theorem 4.1 in (Kingma & Ba, 2015) which provides a claim of convergence for ADAM. As our examples showing non-convergence of ADAM indicate, the proof in (Kingma & Ba, 2015) has problems. The main issue in their proof is the incorrect assumption that $\Gamma_t$ defined in their equation (3) is positive semidefinite, and we also identified problems in lemmas 10.3 and 10.4 in their paper. The following proof fixes these issues and provides a proof of convergence for AMSGRAD.

*Proof.* We begin with the following observation:

$$x_{t+1} = \Pi_{\mathcal{F},\sqrt{\hat{V}_t}}(x_t - \alpha_t \hat{V}_t^{-1/2} m_t) = \min_{x \in \mathcal{F}} \|\hat{V}_t^{1/4}(x - (x_t - \alpha_t \hat{V}_t^{-1/2} m_t))\|.$$

Furthermore, $\Pi_{\mathcal{F},\sqrt{\hat{V}_t}}(x^*) = x^*$ for all $x^* \in \mathcal{F}$. In this proof, we will use $x_i^*$ to denote the $i^{\text{th}}$ coordinate of $x^*$. Using Lemma 4 with $u_1 = x_{t+1}$ and $u_2 = x^*$, we have the following:

$$\|\hat{V}_t^{1/4}(x_{t+1} - x^*)\|^2 \leq \|\hat{V}_t^{1/4}(x_t - \alpha_t \hat{V}_t^{-1/2} m_t - x^*)\|^2$$
$$= \|\hat{V}_t^{1/4}(x_t - x^*)\|^2 + \alpha_t^2 \|\hat{V}_t^{-1/4} m_t\|^2 - 2\alpha_t\langle m_t, x_t - x^*\rangle$$
$$= \|\hat{V}_t^{1/4}(x_t - x^*)\|^2 + \alpha_t^2 \|\hat{V}_t^{-1/4} m_t\|^2 - 2\alpha_t\langle \beta_{1t} m_{t-1} + (1-\beta_{1t})g_t, x_t - x^*\rangle$$

Rearranging the above inequality, we have

$$\langle g_t, x_t - x^*\rangle \leq \frac{1}{2\alpha_t(1-\beta_{1t})}\left[\|\hat{V}_t^{1/4}(x_t - x^*)\|^2 - \|\hat{V}_t^{1/4}(x_{t+1} - x^*)\|^2\right] + \frac{\alpha_t}{2(1-\beta_{1t})}\|\hat{V}_t^{-1/4} m_t\|^2$$
$$+ \frac{\beta_{1t}}{1-\beta_{1t}}\langle m_{t-1}, x_t - x^*\rangle$$
$$\leq \frac{1}{2\alpha_t(1-\beta_{1t})}\left[\|\hat{V}_t^{1/4}(x_t - x^*)\|^2 - \|\hat{V}_t^{1/4}(x_{t+1} - x^*)\|^2\right] + \frac{\alpha_t}{2(1-\beta_{1t})}\|\hat{V}_t^{-1/4} m_t\|^2$$
$$+ \frac{\beta_{1t}}{2(1-\beta_{1t})}\alpha_t\|\hat{V}_t^{-1/4} m_{t-1}\|^2 + \frac{\beta_{1t}}{2\alpha_t(1-\beta_{1t})}\|\hat{V}_t^{1/4}(x_t - x^*)\|^2.$$
$$(15)$$

The second inequality follows from simple application of Cauchy-Schwarz and Young's inequality. We now use the standard approach of bounding the regret at each step using convexity of the function

---

[2]Projections can be easily handled with a little bit of work but the analysis becomes more messy.

$f_t$ in the following manner:

$$\sum_{t=1}^{T} f_t(x_t) - f_t(x^*) \leq \sum_{t=1}^{T} \langle g_t, x_t - x^* \rangle$$

$$\leq \sum_{t=1}^{T} \left[ \frac{1}{2\alpha_t(1-\beta_{1t})} \left[ \|\hat{V}_t^{1/4}(x_t - x^*)\|^2 - \|\hat{V}_t^{1/4}(x_{t+1} - x^*)\|^2 \right] + \frac{\alpha_t}{2(1-\beta_{1t})} \|\hat{V}_t^{-1/4}m_t\|^2 \right.$$

$$\left. + \frac{\beta_{1t}}{2(1-\beta_{1t})}\alpha_t\|\hat{V}_t^{-1/4}m_{t-1}\|^2 + \frac{\beta_{1t}}{2\alpha_t(1-\beta_{1t})}\|\hat{V}_t^{1/4}(x_t - x^*)\|^2 \right].$$

$$(16)$$

The first inequality is due to convexity of function $f_t$. The second inequality follows from the bound in Equation (15). For further bounding this inequality, we need the following intermediate result.

**Lemma 2.** *For the parameter settings and conditions assumed in Theorem 4, we have*

$$\sum_{t=1}^{T} \alpha_t\|\hat{V}_t^{-1/4}m_t\|^2 \leq \frac{\alpha\sqrt{1+\log T}}{(1-\beta_1)(1-\gamma)\sqrt{(1-\beta_2)}} \sum_{i=1}^{d} \|g_{1:T,i}\|_2$$

*Proof.* We start with the following:

$$\sum_{t=1}^{T} \alpha_t\|\hat{V}_t^{-1/4}m_t\|^2 = \sum_{t=1}^{T-1} \alpha_t\|\hat{V}_t^{-1/4}m_t\|^2 + \alpha_T \sum_{i=1}^{d} \frac{m_{T,i}^2}{\sqrt{\hat{v}_{T,i}}}$$

$$\leq \sum_{t=1}^{T-1} \alpha_t\|\hat{V}_t^{-1/4}m_t\|^2 + \alpha_T \sum_{i=1}^{d} \frac{m_{T,i}^2}{\sqrt{v_{T,i}}}$$

$$\leq \sum_{t=1}^{T-1} \alpha_t\|\hat{V}_t^{-1/4}m_t\|^2 + \alpha \sum_{i=1}^{d} \frac{(\sum_{j=1}^{T}(1-\beta_{1j})\Pi_{k=1}^{T-j}\beta_{1(T-k+1)}g_{j,i})^2}{\sqrt{T((1-\beta_2)\sum_{j=1}^{T}\beta_2^{T-j}g_{j,i}^2)}}$$

The first inequality follows from the definition of $\hat{v}_{T,i}$, which is maximum of all $v_{T,i}$ until the current time step. The second inequality follows from the update rule of Algorithm 2. We further bound the above inequality in the following manner:

$$\sum_{t=1}^{T} \alpha_t\|\hat{V}_t^{-1/4}m_t\|^2 \leq \sum_{t=1}^{T-1} \alpha_t\|\hat{V}_t^{-1/4}m_t\|^2 + \alpha \sum_{i=1}^{d} \frac{(\sum_{j=1}^{T}\Pi_{k=1}^{T-j}\beta_{1(T-k+1)})(\sum_{j=1}^{T}\Pi_{k=1}^{T-j}\beta_{1(T-k+1)}g_{j,i}^2)}{\sqrt{T((1-\beta_2)\sum_{j=1}^{T}\beta_2^{T-j}g_{j,i}^2)}}$$

$$\leq \sum_{t=1}^{T-1} \alpha_t\|\hat{V}_t^{-1/4}m_t\|^2 + \alpha \sum_{i=1}^{d} \frac{(\sum_{j=1}^{T}\beta_1^{T-j})(\sum_{j=1}^{T}\beta_1^{T-j}g_{j,i}^2)}{\sqrt{T((1-\beta_2)\sum_{j=1}^{T}\beta_2^{T-j}g_{j,i}^2)}}$$

$$\leq \sum_{t=1}^{T-1} \alpha_t\|\hat{V}_t^{-1/4}m_t\|^2 + \frac{\alpha}{1-\beta_1} \sum_{i=1}^{d} \frac{\sum_{j=1}^{T}\beta_1^{T-j}g_{j,i}^2}{\sqrt{T((1-\beta_2)\sum_{j=1}^{T}\beta_2^{T-j}g_{j,i}^2)}}$$

$$\leq \sum_{t=1}^{T-1} \alpha_t\|\hat{V}_t^{-1/4}m_t\|^2 + \frac{\alpha}{(1-\beta_1)\sqrt{T(1-\beta_2)}} \sum_{i=1}^{d}\sum_{j=1}^{T} \frac{\beta_1^{T-j}g_{j,i}^2}{\sqrt{\beta_2^{T-j}g_{j,i}^2}}$$

$$\leq \sum_{t=1}^{T-1} \alpha_t\|\hat{V}_t^{-1/4}m_t\|^2 + \frac{\alpha}{(1-\beta_1)\sqrt{T(1-\beta_2)}} \sum_{i=1}^{d}\sum_{j=1}^{T} \gamma^{T-j}|g_{j,i}| \qquad (17)$$

The first inequality follows from Cauchy-Schwarz inequality. The second inequality is due to the fact that $\beta_{1k} \leq \beta_1$ for all $k \in [T]$. The third inequality follows from the inequality $\sum_{j=1}^{T}\beta_1^{T-j} \leq 1/(1-\beta_1)$. By using similar upper bounds for all time steps, the quantity in Equation (17) can

further be bounded as follows:

$$\sum_{t=1}^{T} \alpha_t \|\hat{V}_t^{-1/4} m_t\|^2 \le \sum_{t=1}^{T} \frac{\alpha}{(1-\beta_1)\sqrt{t(1-\beta_2)}} \sum_{i=1}^{d} \sum_{j=1}^{t} \gamma^{t-j} |g_{j,i}|$$

$$= \frac{\alpha}{(1-\beta_1)\sqrt{(1-\beta_2)}} \sum_{i=1}^{d} \sum_{t=1}^{T} \frac{1}{\sqrt{t}} \sum_{j=1}^{t} \gamma^{t-j} |g_{j,i}| = \frac{\alpha}{(1-\beta_1)\sqrt{(1-\beta_2)}} \sum_{i=1}^{d} \sum_{t=1}^{T} |g_{t,i}| \sum_{j=t}^{T} \frac{\gamma^{j-t}}{\sqrt{j}}$$

$$\le \frac{\alpha}{(1-\beta_1)\sqrt{(1-\beta_2)}} \sum_{i=1}^{d} \sum_{t=1}^{T} |g_{t,i}| \sum_{j=t}^{T} \frac{\gamma^{j-t}}{\sqrt{t}} \le \frac{\alpha}{(1-\beta_1)\sqrt{(1-\beta_2)}} \sum_{i=1}^{d} \sum_{t=1}^{T} |g_{t,i}| \frac{1}{(1-\gamma)\sqrt{t}}$$

$$\le \frac{\alpha}{(1-\beta_1)(1-\gamma)\sqrt{(1-\beta_2)}} \sum_{i=1}^{d} \|g_{1:T,i}\|_2 \sqrt{\sum_{t=1}^{T} \frac{1}{t}} \le \frac{\alpha\sqrt{1+\log T}}{(1-\beta_1)(1-\gamma)\sqrt{(1-\beta_2)}} \sum_{i=1}^{d} \|g_{1:T,i}\|_2$$

The third inequality follows from the fact that $\sum_{j=t}^{T} \gamma^{j-t} \le 1/(1-\gamma)$. The fourth inequality is due to simple application of Cauchy-Schwarz inequality. The final inequality is due to the following bound on harmonic sum: $\sum_{t=1}^{T} 1/t \le (1 + \log T)$. This completes the proof of the lemma.

We now return to the proof of Theorem 4. Using the above lemma in Equation (16) , we have:

$$\sum_{t=1}^{T} f_t(x_t) - f_t(x^*) \le \sum_{t=1}^{T} \left[ \frac{1}{2\alpha_t(1-\beta_{1t})} \left[ \|\hat{V}_t^{1/4}(x_t - x^*)\|^2 - \|\hat{V}_t^{1/4}(x_{t+1} - x^*)\|^2 \right] \right.$$

$$\left. + \frac{\beta_{1t}}{2\alpha_t(1-\beta_{1t})} \|\hat{V}_t^{1/4}(x_t - x^*)\|^2 \right] + \frac{\alpha\sqrt{1+\log T}}{(1-\beta_1)^2(1-\gamma)\sqrt{(1-\beta_2)}} \sum_{i=1}^{d} \|g_{1:T,i}\|_2$$

$$\le \frac{1}{2\alpha_1(1-\beta_1)} \|\hat{V}_1^{1/4}(x_1 - x^*)\|^2 + \frac{1}{2(1-\beta_1)} \sum_{t=2}^{T} \left[ \frac{\|\hat{V}_t^{1/4}(x_t - x^*)\|^2}{\alpha_t} - \frac{\|\hat{V}_{t-1}^{1/4}(x_t - x^*)\|^2}{\alpha_{t-1}} \right]$$

$$+ \sum_{t=1}^{T} \left[ \frac{\beta_{1t}}{2\alpha_t(1-\beta_1)} \|\hat{V}_t^{1/4}(x_t - x^*)\|^2 \right] + \frac{\alpha\sqrt{1+\log T}}{(1-\beta_1)^2(1-\gamma)\sqrt{(1-\beta_2)}} \sum_{i=1}^{d} \|g_{1:T,i}\|_2$$

$$= \frac{1}{2\alpha_1(1-\beta_1)} \sum_{i=1}^{d} \hat{v}_{1,i}^{1/2} (x_{1,i} - x_i^*)^2 + \frac{1}{2(1-\beta_1)} \sum_{t=2}^{T} \sum_{i=1}^{d} (x_{t,i} - x_i^*)^2 \left[ \frac{\hat{v}_{t,i}^{1/2}}{\alpha_t} - \frac{\hat{v}_{t-1,i}^{1/2}}{\alpha_{t-1}} \right]$$

$$+ \frac{1}{2(1-\beta_1)} \sum_{t=1}^{T} \sum_{i=1}^{d} \frac{\beta_{1t}(x_{t,i} - x_i^*)^2 \hat{v}_{t,i}^{1/2}}{\alpha_t} + \frac{\alpha\sqrt{1+\log T}}{(1-\beta_1)^2(1-\gamma)\sqrt{(1-\beta_2)}} \sum_{i=1}^{d} \|g_{1:T,i}\|_2.$$

$$(18)$$

The first inequality and second inequality use the fact that $\beta_{1t} \le \beta_1$. In order to further simplify the bound in Equation (18), we need to use telescopic sum. We observe that, by definition of $\hat{v}_{t,i}$, we have

$$\frac{\hat{v}_{t,i}^{1/2}}{\alpha_t} \ge \frac{\hat{v}_{t-1,i}^{1/2}}{\alpha_{t-1}}.$$

Using the $L_\infty$ bound on the feasible region and making use of the above property in Equation (18), we have:

$$\sum_{t=1}^{T} f_t(x_t) - f_t(x^*) \le \frac{1}{2\alpha_1(1-\beta_1)} \sum_{i=1}^{d} \hat{v}_{1,i}^{1/2} D_\infty^2 + \frac{1}{2(1-\beta_1)} \sum_{t=2}^{T} \sum_{i=1}^{d} D_\infty^2 \left[ \frac{\hat{v}_{t,i}^{1/2}}{\alpha_t} - \frac{\hat{v}_{t-1,i}^{1/2}}{\alpha_{t-1}} \right]$$

$$+ \frac{1}{2(1-\beta_1)} \sum_{t=1}^{T} \sum_{i=1}^{d} \frac{D_\infty^2 \beta_{1t} \hat{v}_{t,i}^{1/2}}{\alpha_t} + \frac{\alpha\sqrt{1+\log T}}{(1-\beta_1)^2(1-\gamma)\sqrt{(1-\beta_2)}} \sum_{i=1}^{d} \|g_{1:T,i}\|_2$$

$$= \frac{D_\infty^2}{2\alpha_T(1-\beta_1)} \sum_{i=1}^{d} \hat{v}_{T,i}^{1/2} + \frac{D_\infty^2}{2(1-\beta_1)} \sum_{t=1}^{T} \sum_{i=1}^{d} \frac{\beta_{1t} \hat{v}_{t,i}^{1/2}}{\alpha_t} + \frac{\alpha\sqrt{1+\log T}}{(1-\beta_1)^2(1-\gamma)\sqrt{(1-\beta_2)}} \sum_{i=1}^{d} \|g_{1:T,i}\|_2.$$

---

**Algorithm 3** ADAMNC

  **Input:** $x_1 \in \mathcal{F}$, step size $\{\alpha_t > 0\}_{t=1}^T$, $\{(\beta_{1t}, \beta_{2t})\}_{t=1}^T$
  Set $m_0 = 0$ and $v_0 = 0$
  **for** $t = 1$ **to** $T$ **do**
    $g_t = \nabla f_t(x_t)$
    $m_t = \beta_{1t} m_{t-1} + (1 - \beta_{1t}) g_t$
    $v_t = \beta_{2t} v_{t-1} + (1 - \beta_{2t}) g_t^2$ and $V_t = \text{diag}(v_t)$
    $x_{t+1} = \Pi_{\mathcal{F}, \sqrt{V_t}}(x_t - \alpha_t m_t / \sqrt{v_t})$
  **end for**

---

The equality follows from simple telescopic sum, which yields the desired result. One important point to note here is that the regret of AMSGRAD can be bounded by $O(G_\infty \sqrt{T})$. This can be easily seen from the proof of the aforementioned lemma where in the analysis the term $\sum_{t=1}^T |g_{t,i}|/\sqrt{t}$ can also be bounded by $O(G_\infty \sqrt{T})$. Thus, the regret of AMSGRAD is upper bounded by minimum of $O(G_\infty \sqrt{T})$ and the bound in the Theorem 4 and therefore, the worst case dependence of regret on $T$ in our case is $O(\sqrt{T})$. $\qquad\square$

## E  PROOF OF THEOREM 5

*Proof.* Using similar argument to proof of Theorem 4 until Equation (15), we have the following

$$\langle g_t, x_t - x^* \rangle \leq \frac{1}{2\alpha_t(1 - \beta_{1t})} \left[ \|V_t^{1/4}(x_t - x^*)\|^2 - \|V_t^{1/4}(x_{t+1} - x^*)\|^2 \right] + \frac{\alpha_t}{2(1 - \beta_{1t})} \|V_t^{-1/4} m_t\|^2$$

$$+ \frac{\beta_{1t}}{2(1 - \beta_{1t})} \alpha_t \|V_t^{-1/4} m_{t-1}\|^2 + \frac{\beta_{1t}}{2\alpha_t(1 - \beta_{1t})} \|V_t^{1/4}(x_t - x^*)\|^2. \tag{19}$$

The second inequality follows from simple application of Cauchy-Schwarz and Young's inequality. We now use the standard approach of bounding the regret at each step using convexity of the function $f_t$ in the following manner:

$$\sum_{t=1}^T f_t(x_t) - f_t(x^*) \leq \sum_{t=1}^T \langle g_t, x_t - x^* \rangle$$

$$\leq \sum_{t=1}^T \left[ \frac{1}{2\alpha_t(1 - \beta_{1t})} \left[ \|V_t^{1/4}(x_t - x^*)\|^2 - \|V_t^{1/4}(x_{t+1} - x^*)\|^2 \right] + \frac{\alpha_t}{2(1 - \beta_{1t})} \|V_t^{-1/4} m_t\|^2 \right.$$

$$\left. + \frac{\beta_{1t}}{2(1 - \beta_{1t})} \alpha_t \|V_t^{-1/4} m_{t-1}\|^2 + \frac{\beta_{1t}}{2\alpha_t(1 - \beta_{1t})} \|V_t^{1/4}(x_t - x^*)\|^2 \right]. \tag{20}$$

The inequalities follow due to convexity of function $f_t$ and Equation (19). For further bounding this inequality, we need the following intermediate result.

**Lemma 3.** *For the parameter settings and conditions assumed in Theorem 5, we have*

$$\sum_{t=1}^T \alpha_t \|V_t^{-1/4} m_t\|^2 \leq \frac{2\zeta}{(1 - \beta_1)^2} \sum_{i=1}^d \|g_{1:T,i}\|_2.$$

*Proof.* We start with the following:

$$\sum_{t=1}^T \alpha_t \|V_t^{-1/4} m_t\|^2 = \sum_{t=1}^{T-1} \alpha_t \|V_t^{-1/4} m_t\|^2 + \alpha_T \sum_{i=1}^d \frac{m_{T,i}^2}{\sqrt{v_{T,i}}}$$

$$\leq \sum_{t=1}^{T-1} \alpha_t \|\hat{V}_t^{-1/4} m_t\|^2 + \alpha_T \sum_{i=1}^d \frac{(\sum_{j=1}^T (1 - \beta_{1j}) \Pi_{k=1}^{T-j} \beta_{1(T-k+1)} g_{j,i})^2}{\sqrt{(\sum_{j=1}^T \Pi_{k=1}^{T-j} \beta_{2(T-k+1)} (1 - \beta_{2j}) g_{j,i}^2)}}$$

The first inequality follows from the update rule of Algorithm 2. We further bound the above inequality in the following manner:

$$
\begin{aligned}
\sum_{t=1}^{T} \alpha_t \|V_t^{-1/4} m_t\|^2 &\leq \sum_{t=1}^{T-1} \alpha_t \|V_t^{-1/4} m_t\|^2 + \alpha_T \sum_{i=1}^{d} \frac{(\sum_{j=1}^{T} \Pi_{k=1}^{T-j} \beta_{1(T-k+1)})(\sum_{j=1}^{T} \Pi_{k=1}^{T-j} \beta_{1(T-k+1)} g_{j,i}^2)}{\sqrt{\sum_{j=1}^{T} \Pi_{k=1}^{T-j} \beta_{2(T-k+1)}(1-\beta_{2j}) g_{j,i}^2}} \\
&\leq \sum_{t=1}^{T-1} \alpha_t \|V_t^{-1/4} m_t\|^2 + \alpha_T \sum_{i=1}^{d} \frac{(\sum_{j=1}^{T} \beta_1^{T-j})(\sum_{j=1}^{T} \beta_1^{T-j} g_{j,i}^2)}{\sqrt{\sum_{j=1}^{T} \Pi_{k=1}^{T-j} \beta_{2(T-k+1)}(1-\beta_{2j}) g_{j,i}^2}} \\
&\leq \sum_{t=1}^{T-1} \alpha_t \|V_t^{-1/4} m_t\|^2 + \frac{\alpha_T}{1-\beta_1} \sum_{i=1}^{d} \frac{\sum_{j=1}^{T} \beta_1^{T-j} g_{j,i}^2}{\sqrt{\sum_{j=1}^{T} \Pi_{k=1}^{T-j} \beta_{2(T-k+1)}(1-\beta_{2j}) g_{j,i}^2}} \\
&\leq \sum_{t=1}^{T-1} \alpha_t \|V_t^{-1/4} m_t\|^2 + \frac{\zeta}{1-\beta_1} \sum_{i=1}^{d} \frac{\sum_{j=1}^{T} \beta_1^{T-j} g_{j,i}^2}{\sqrt{\sum_{j=1}^{T} g_{j,i}^2}} \\
&\leq \sum_{t=1}^{T-1} \alpha_t \|V_t^{-1/4} m_t\|^2 + \frac{\zeta}{1-\beta_1} \sum_{i=1}^{d} \sum_{j=1}^{T} \frac{\beta_1^{T-j} g_{j,i}^2}{\sqrt{\sum_{k=1}^{j} g_{k,i}^2}}
\end{aligned}
\tag{21}
$$

The first inequality follows from Cauchy-Schwarz inequality. The second inequality is due to the fact that $\beta_{1k} \leq \beta_1$ for all $k \in [T]$. The third inequality follows from the inequality $\sum_{j=1}^{T} \beta_1^{T-j} \leq 1/(1-\beta_1)$. Using similar argument for all time steps, the quantity in Equation (21) can be bounded as follows:

$$
\begin{aligned}
\sum_{t=1}^{T} \alpha_t \|V_t^{-1/4} m_t\|^2 &\leq \frac{\zeta}{1-\beta_1} \sum_{i=1}^{d} \sum_{j=1}^{T} \frac{\sum_{l=0}^{T-j} \beta_1^l g_{j,i}^2}{\sqrt{\sum_{k=1}^{j} g_{k,i}^2}} \\
&\leq \frac{\zeta}{(1-\beta_1)^2} \sum_{i=1}^{d} \sum_{j=1}^{T} \frac{g_{j,i}^2}{\sqrt{\sum_{k=1}^{j} g_{k,i}^2}} \leq \frac{2\zeta}{(1-\beta_1)^2} \sum_{i=1}^{d} \|g_{1:T,i}\|_2.
\end{aligned}
$$

The final inequality is due to Lemma 5. This completes the proof of the lemma.

Using the above lemma in Equation (20) , we have:

$$
\begin{aligned}
\sum_{t=1}^{T} f_t(x_t) - f_t(x^*) &\leq \sum_{t=1}^{T} \left[ \frac{1}{2\alpha_1(1-\beta_{1t})} \left[ \|V_t^{1/4}(x_t - x^*)\|^2 - \|V_t^{1/4}(x_{t+1} - x^*)\|^2 \right] \right. \\
&\quad \left. + \frac{\beta_{1t}}{2\alpha_t(1-\beta_{1t})} \|V_t^{1/4}(x_t - x^*)\|^2 \right] + \frac{2\zeta}{(1-\beta_1)^3} \sum_{i=1}^{d} \|g_{1:T,i}\|_2 \\
&\leq \frac{1}{2\alpha_1(1-\beta_1)} \|V_1^{1/4}(x_1 - x^*)\|^2 + \frac{1}{2(1-\beta_1)} \sum_{t=2}^{T} \left[ \frac{\|V_t^{1/4}(x_t - x^*)\|^2}{\alpha_t} - \frac{\|V_{t-1}^{1/4}(x_{t-1} - x^*)\|^2}{\alpha_t} \right] \\
&\quad + \sum_{t=1}^{T} \left[ \frac{\beta_{1t}}{2\alpha_t(1-\beta_1)} \|V_t^{1/4}(x_t - x^*)\|^2 \right] + \frac{2\zeta}{(1-\beta_1)^3} \sum_{i=1}^{d} \|g_{1:T,i}\|_2 \\
&= \frac{1}{2\alpha_1(1-\beta_1)} \sum_{i=1}^{d} v_{1,i}^{1/2}(x_{1,i} - x_i^*)^2 + \frac{1}{2(1-\beta_1)} \sum_{t=2}^{T} \sum_{i=1}^{d} (x_{t,i} - x_i^*)^2 \left[ \frac{v_{t,i}^{1/2}}{\alpha_t} - \frac{v_{t-1,i}^{1/2}}{\alpha_{t-1}} \right] \\
&\quad + \frac{1}{2(1-\beta_1)} \sum_{t=1}^{T} \sum_{i=1}^{d} \frac{\beta_{1t}(x_{t,i} - x_i^*)^2 v_{t,i}^{1/2}}{\alpha_t} + \frac{2\zeta}{(1-\beta_1)^3} \sum_{i=1}^{d} \|g_{1:T,i}\|_2.
\end{aligned}
\tag{22}
$$

The first inequality and second inequality use the fact that $\beta_{1t} \leq \beta_1$. Furthermore, from the theorem statement, we know that that $\{(\alpha_t . \beta_{2t})\}$ are selected such that the following holds:

$$
\frac{v_{t,i}^{1/2}}{\alpha_t} \geq \frac{v_{t-1,i}^{1/2}}{\alpha_{t-1}}.
$$

Using the $L_\infty$ bound on the feasible region and making use of the above property in Equation (22), we have:

$$\sum_{t=1}^{T} f_t(x_t) - f_t(x^*) \leq \frac{1}{2\alpha_1(1-\beta_1)} \sum_{i=1}^{d} v_{1,i}^{1/2} D_\infty^2 + \frac{1}{2(1-\beta_1)} \sum_{t=2}^{T} \sum_{i=1}^{d} D_\infty^2 \left[ \frac{v_{t,i}^{1/2}}{\alpha_t} - \frac{v_{t-1,i}^{1/2}}{\alpha_{t-1}} \right]$$

$$+ \frac{1}{2(1-\beta_1)} \sum_{t=1}^{T} \sum_{i=1}^{d} \frac{D_\infty^2 \beta_{1t} v_{t,i}^{1/2}}{\alpha_t} + \frac{2\zeta}{(1-\beta_1)^3} \sum_{i=1}^{d} \|g_{1:T,i}\|_2$$

$$= \frac{D_\infty^2}{2\alpha_T(1-\beta_1)} \sum_{i=1}^{d} v_{T,i}^{1/2} + \frac{D_\infty^2}{2(1-\beta_1)} \sum_{t=1}^{T} \sum_{i=1}^{d} \frac{\beta_{1t} v_{t,i}^{1/2}}{\alpha_t} + \frac{2\zeta}{(1-\beta_1)^3} \sum_{i=1}^{d} \|g_{1:T,i}\|_2.$$

The equality follows from simple telescopic sum, which yields the desired result. □

## F   PROOF OF THEOREM 6

**Theorem 6.** *For any $\epsilon > 0$, ADAM with the modified update in Equation (3) and with parameter setting such that all the conditions in (Kingma & Ba, 2015) are satisfied can have non-zero average regret i.e., $R_T/T \nrightarrow 0$ as $T \to \infty$ for convex $\{f_i\}_{i=1}^{\infty}$ with bounded gradients on a feasible set $\mathcal{F}$ having bounded $D_\infty$ diameter.*

*Proof.* Let us first consider the case where $\epsilon = 1$ (in fact, the same setting works for any $\epsilon \leq 1$). The general $\epsilon$ case can be proved by simply rescaling the sequence of functions by a factor of $\sqrt{\epsilon}$. We show that the same optimization setting in Theorem 1 where $f_t$ are linear functions and $\mathcal{F} = [-1, 1]$, hence, we only discuss the details that differ from the proof of Theorem 1. In particular, we define the following function sequence:

$$f_t(x) = \begin{cases} Cx, & \text{for } t \bmod 3 = 1 \\ -x, & \text{otherwise,} \end{cases}$$

where $C \geq 2$. Similar to the proof of Theorem 1, we assume that the initial point is $x_1 = 1$ and the parameters are:

$$\beta_1 = 0, \beta_2 = \frac{2}{(1+C^2)C^2} \text{ and } \alpha_t = \frac{\alpha}{\sqrt{t}}$$

where $\alpha < \sqrt{1-\beta_2}$. The proof essentially follows along the lines of that of Theorem 1 and is through principle of mathematical induction. Our aim is to prove that $x_{3t+2}$ and $x_{3t+3}$ are positive and $x_{3t+4} = 1$. The base case holds trivially. Suppose for some $t \in \mathbb{N} \cup \{0\}$, we have $x_i > 0$ for all $i \in [3t+1]$ and $x_{3t+1} = 1$. For $(3t+1)^{\text{th}}$ update, the only change from the update of in Equation (1) is the additional $\epsilon$ in the denominator i.e., we have

$$\hat{x}_{3t+2} = x_{3t+1} - \frac{\alpha C}{\sqrt{(3t+1)(\beta_2 v_{3t} + (1-\beta_2)C^2 + \epsilon)}}$$

$$\geq 1 - \frac{\alpha C}{\sqrt{(3t+1)(\beta_2 v_{3t} + (1-\beta_2)C^2)}} \geq 0.$$

The last inequality follows by simply dropping $v_{3t}$ term and using the relation that $\alpha < \sqrt{1-\beta_2}$. Therefore, we have $0 < \hat{x}_{3t+2} < 1$ and hence $x_{3t+2} = \hat{x}_{3t+2} > 0$. Furthermore, after the $(3t+2)^{\text{th}}$ and $(3t+3)^{\text{th}}$ updates of ADAM in Equation (1), we have the following:

$$\hat{x}_{3t+3} = x_{3t+2} + \frac{\alpha}{\sqrt{(3t+2)(\beta_2 v_{3t+1} + (1-\beta_2) + \epsilon)}},$$

$$\hat{x}_{3t+4} = x_{3t+3} + \frac{\alpha}{\sqrt{(3t+3)(\beta_2 v_{3t+2} + (1-\beta_2) + \epsilon)}}.$$

Since $x_{3t+2} > 0$, it is easy to see that $x_{3t+3} > 0$. To complete the proof, we need to show that $x_{3t+4} = 1$. The only change here from the proof of Theorem 1 is that we need to show the

following:

$$
\frac{\alpha}{\sqrt{(3t+2)(\beta_2 v_{3t+1} + (1-\beta_2) + \epsilon)}} + \frac{\alpha}{\sqrt{(3t+3)(\beta_2 v_{3t+2} + (1-\beta_2) + \epsilon)}}
$$

$$
\geq \frac{\alpha}{\sqrt{\beta_2 C^2 + (1-\beta_2) + \epsilon}} \left( \frac{1}{\sqrt{3t+2}} + \frac{1}{\sqrt{3t+3}} \right)
$$

$$
\geq \frac{\alpha}{\sqrt{\beta_2 C^2 + (1-\beta_2) + \epsilon}} \left( \frac{1}{\sqrt{2(3t+1)}} + \frac{1}{\sqrt{2(3t+1)}} \right)
$$

$$
= \frac{\sqrt{2}\alpha}{\sqrt{(3t+1)(\beta_2 C^2 + (1-\beta_2) + \epsilon)}} = \frac{\alpha C}{\sqrt{(3t+1)((1-\beta_2)C^2 + \epsilon)}}
$$

$$
\geq \frac{\alpha C}{\sqrt{(3t+1)(\beta_2 v_{3t} + (1-\beta_2)C^2 + \epsilon)}}. \tag{23}
$$

The first inequality is due to the fact that $v_t \leq C^2$ for all $t \in \mathbb{N}$. The last equality is due to following fact:

$$
\sqrt{\frac{\beta_2 C^2 + (1-\beta_2)}{2}} = \sqrt{1 - \beta_2 + \frac{\epsilon}{C^2}}.
$$

for the choice of $\beta_2 = 2/[(1+C^2)C^2]$ and $\epsilon = 1$. Therefore, we see that $x_{3t+4} = 1$. Therefore, by the principle of mathematical induction it holds for all $t \in \mathbb{N} \cup \{0\}$. Thus, we have

$$
f_{3t+1}(x_{3t+1}) + f_{3t+2}(x_{3t+2}) + f_{3t+2}(x_{3t+2}) - f_{3t+1}(-1) - f_{3t+2}(-1) - f_{3t+3}(-1) \geq 2C - 4.
$$

Therefore, for every 3 steps, ADAM suffers a regret of at least $2C - 4$. More specifically, $R_T \geq (2C-4)T/3$. Since $C \geq 2$, this regret can be very large and furthermore, $R_T/T \nrightarrow 0$ as $T \to \infty$, which completes the proof of the case where $\epsilon = 1$. For the general $\epsilon$ case, we consider the following sequence of functions:

$$
f_t(x) = \begin{cases} C\sqrt{\epsilon}x, & \text{for } t \bmod 3 = 1 \\ -\sqrt{\epsilon}x, & \text{otherwise,} \end{cases}
$$

The functions are essentially rescaled in a manner so that the resultant updates of ADAM correspond to the one in the optimization setting described above. Using essentially the same argument as above, it is easy to show that the regret $R_T \geq (2C-4)\sqrt{\epsilon}T/3$ and thus, the average regret is non-zero asymptotically, which completes the proof. $\qquad \square$

## G   AUXILIARY LEMMA

**Lemma 4** ((McMahan & Streeter, 2010)). *For any $Q \in \mathcal{S}_+^d$ and convex feasible set $\mathcal{F} \subset \mathbb{R}^d$, suppose $u_1 = \min_{x \in \mathcal{F}} \|Q^{1/2}(x - z_1)\|$ and $u_2 = \min_{x \in \mathcal{F}} \|Q^{1/2}(x - z_2)\|$ then we have $\|Q^{1/2}(u_1 - u_2)\| \leq \|Q^{1/2}(z_1 - z_2)\|$.*

*Proof.* We provide the proof here for completeness. Since $u_1 = \min_{x \in \mathcal{F}} \|Q^{1/2}(x - z_1)\|$ and $u_2 = \min_{x \in \mathcal{F}} \|Q^{1/2}(x - z_2)\|$ and from the property of projection operator we have the following:

$$
\langle z_1 - u_1, Q(z_2 - z_1) \rangle \geq 0 \text{ and } \langle z_2 - u_2, Q(z_1 - z_2) \rangle \geq 0.
$$

Combining the above inequalities, we have

$$
\langle u_2 - u_1, Q(z_2 - z_1) \rangle \geq \langle z_2 - z_1, Q(z_2 - z_1) \rangle. \tag{24}
$$

Also, observe the following:

$$
\langle u_2 - u_1, Q(z_2 - z_1) \rangle \leq \frac{1}{2} [\langle u_2 - u_1, Q(u_2 - u_1) \rangle + \langle z_2 - z_1, Q(z_2 - z_1) \rangle]
$$

The above inequality can be obtained from the fact that $\langle (u_2 - u_1) - (z_2 - z_1), Q((u_2 - u_1) - (z_2 - z_1)) \rangle \geq 0$ as $Q \in \mathcal{S}_+^d$ and rearranging the terms. Combining the above inequality with Equation (24), we have the required result. $\qquad \square$

**Lemma 5** ((Auer & Gentile, 2000)). *For any non-negative real numbers $y_1, \cdots, y_t$, the following holds:*

$$\sum_{i=1}^{t} \frac{y_i}{\sqrt{\sum_{j=1}^{i} y_j}} \leq 2\sqrt{\sum_{i=1}^{t} y_i}.$$

**Lemma 6.** *Suppose $\mathcal{F} = [a, b]$ for $a, b \in \mathbb{R}$ and*

$$y_{t+1} = \Pi_{\mathcal{F}}(y_t + \delta_t)$$

*for all the $t \in [T]$, $y_1 \in \mathcal{F}$ and furthermore, there exists $i \in [T]$ such that $\delta_j \leq 0$ for all $j \leq i$ and $\delta_j > 0$ for all $j > i$. Then we have,*

$$y_{T+1} \geq \min\{b, y_1 + \sum_{j=1}^{T} \delta_j\}.$$

*Proof.* It is first easy to see that $y_{i+1} \geq y_1 + \sum_{j=1}^{i} \delta_j$ since $\delta_j \leq 0$ for all $j \leq i$. Furthermore, also observe that $y_{T+1} \geq \min\{b, y_{i+1} + \sum_{j=i+1}^{T} \delta_j\}$ since $\delta_j \geq 0$ for all $j > i$. Combining the above two inequalities gives us the desired result. □

