# OpenReview forum: "On the Convergence of Adam and Beyond"
_ICLR.cc/2018/Conference — Accept (Oral)_

### Official Review · AnonReviewer1 · 2017-11-23

**Rating:** 9
**Confidence:** 5

**Review:**

The paper presents three contributions: 1) it shows that the proof of convergence Adam is wrong; 2) it presents adversarial and stochastic examples on which Adam converges to the worst possible solution (i.e. there is no hope to just fix Adam's proof); 3) it proposes a variant of Adam called AMSGrad that fixes the problems in the original proof and seems to have good empirical properties.

The contribution of this paper is very relevant to ICLR and, as far as I know, novel.
The result is clearly very important for the deep learning community.
I also checked most of the proofs and they look correct to me: The arguments are quite standard, even if the proofs are very long.

One note on the generality of the results: the papers states that some of the results could apply to RMSProp too. However, it has been proved that RMSProp with a certain settings of its parameters is nothing else than AdaGrad (see Section 4 in  Mukkamala and Hein, ICML'17). Hence, at least for a certain setting of its parameters, RMSProp will converge. Of course, the proof in the ICML paper could be wrong, I did not check that...

A general note on the learning rate: The fact that most of these algorithms are used with a fixed learning rate while the analysis assume a decaying learning rate should hint to the fact that we are not using the right analysis. Indeed, all these variants of AdaGrad did not really improve the AdaGrad's regret bound. In this view, none of these algorithms contributed in any meaningful way to our understanding of the optimization of deep networks *nor* they advanced in any way the state-of-the-art for optimizing convex Lipschitz functions.
On the other hand, analysis of SGD-like algorithms with constant step sizes are known. See, for example, Zhang, ICML'04 where linear convergence is proved in a neighbourhood of the optimal solution for strongly convex problems.
So, even if I understand this is not the main objective of this paper, it would be nice to see a discussion on this point and the limitations of regret analysis to analyse SGD algorithms.

Overall, I strongly suggest to accept this paper.


Suggestions/minor things:
- To facilitate the reader, I would state from the beginning what are the common settings of beta_1 and beta_2 in Adam. This makes easier to see that, for example, the condition of Theorem 2 is verified.
- \hat{v}_{0} is undefined in Algorithm 2.
- The graphs in figure 2 would gain in readability if the setting of each one of them would be added as their titles.
- McMahan and Streeter (2010) is missing the title. (Also, kudos for citing both the independent works on AdaGrad)
- page 11, last equation, 2C-4=2C-4. Same on page 13.
- Lemma 4 contains x_1,x_2,z_1, and z_2: are x_1 and z_1 the same? also x_2 and z_2?

---

> ### Public Comment · (anonymous) · 2017-12-08
> **about your comment on RMSProp for some choice of parameters**
>
> The RMSProp used in Section 4 in Mukkamala and Hein, ICML'17 is not the standard RMSProp but a modification in which the parameter used for computing the geometrical averages of the gradient entries squared changes with time. So there is no contradiction with this paper, that shows counterexamples for the standard algorithm in which that parameter is constant.

---

> ### Author Response · Authors · 2017-12-15
> **Thanks for the very helpful feedback**
>
> We thank the reviewer for very helpful and constructive feedback.
>
> About Mukkamala and Hein 2017 [MH17]: Thanks for pointing this paper. As the anonymous reviewer rightly points out, the [MH17] does not look at the standard version of RMSProp but rather a modification and thus, there is no contradiction with our paper. We will make this point clear in the final version of the paper.
>
> Regarding note about learning rate: While it is true that none of these new rates improve upon Adagrad rates, in fact, in the worst case one cannot improve the regret of standard online gradient descent in general convex setting. Adagrad improves this in the special case of sparse gradients (see for instance, Section 1.3 of Duchi et al. 2011). However, these algorithms, which are designed for specific convex settings, appear to perform reasonably well in the nonconvex settings too (especially in deep networks). Exponential moving average (EMA) variants seem to further improve the performance in the (dense) nonconvex setting. Understanding the cause for good performance in nonconvex settings is an interesting open problem. Our aim was to take an initial step to develop more principled EMA approaches. We will add a description in the final version of the paper.
>
> Lemma 4: Thanks for pointing it out and sorry for the confusion. Indeed, x1 = z1 and x2 = z2. We have corrected this typo.
>
> We have also revised the paper to address the minor typos mentioned in the review.

---

### Official Review · AnonReviewer3 · 2017-11-27
**Noteworthy paper; some (fixable) issues with technical presentation**

**Rating:** 8
**Confidence:** 4

**Review:**

This work identifies a mistake in the existing proof of convergence of
Adam, which is among the most popular optimization methods in deep
learning. Moreover, it gives a simple 1-dimensional counterexample with
linear losses on which Adam does not converge. The same issue also
affects RMSprop, which may be viewed as a special case of Adam without
momentum. The problem with Adam is that the "learning rate" matrices
V_t^{1/2}/alpha_t are not monotonically decreasing. A new method, called
AMSGrad is therefore proposed, which modifies Adam by forcing these
matrices to be decreasing. It is then shown that AMSGrad does satisfy
essentially the same convergence bound as the one previously claimed for
Adam. Experiments and simulations are provided that support the
theoretical analysis.

Apart from some issues with the technical presentation (see below), the
paper is well-written.

Given the popularity of Adam, I consider this paper to make a very
interesting observation. I further believe all issues with the technical
presentation can be readily addressed.



Issues with Technical Presentation:

- All theorems should explicitly state the conditions they require
  instead of referring to "all the conditions in (Kingma & Ba, 2015)".
- Theorem 2 is a repetition of Theorem 1 (except for additional
  conditions).
- The proof of Theorem 3 assumes there are no projections, so this
  should be stated as part of its conditions. (The claim in footnote 2
  that they can be handled seems highly plausible, but you should be up
  front about the limitations of your results.)
- The regret bound Theorem 4 establishes convergence of the optimization
  method, so it plays the role of a sanity check. However, it is
  strictly worse than the regret bound O(sqrt{T}) for online gradient
  descent [Zinkevich,2003], so it cannot explain why the proposed
  AMSgrad method might be adaptive. (The method may indeed be adaptive
  in some sense; I am just saying the *bound* does not express that.
  This is also not a criticism of the current paper; the same remark
  also applies to the previously claimed regret bound for Adam.)
- The discussion following Corollary 1 suggests that sum_i
  hat{v}_{T,i}^{1/2} might be much smaller than d G_infty. This is true,
  but we should always expect it to be at least a constant, because
  hat{v}_{t,i} is monotonically increasing by definition of the
  algorithm, so the bound does not get better than O(sqrt(T)).
  It is also suggested that sum_i ||g_{1:T,i}|| = sqrt{sum_{t=1}^T
  g_{t,i}^2} might be much smaller than dG_infty, but this is very
  unlikely, because this term will typically grow like O(sqrt{T}),
  unless the data are extremely sparse, so we should at least expect
  some dependence on T.
- In the proof of Theorem 1, the initial point is taken to be x_1 = 1,
  which is perfectly fine, but it is not "without loss of generality",
  as claimed. This should be stated in the statement of the Theorem.
- The proof of Theorem 6 in appendix B only covers epsilon=1. If it is
  "easy to show" that the same construction also works for other
  epsilon, as claimed, then please provide the proof for general
  epsilon.


Other remarks:

- Theoretically, nonconvergence of Adam seems a severe problem. Can you
  speculate on why this issue has not prevented its widespread adoption?
  Which factors might mitigate the issue in practice?
- Please define g_t \circ g_t and g_{1:T,i}
- I would recommend sticking with standard linear algebra notation for
  the sqrt and the inverse of a matrix and simply using A^{-1} and
  A^{1/2} instead of 1/A and sqrt{A}.
- In theorems 1,2,3, I would recommend stating the dimension (d=1) of
  your counterexamples, which makes them very nice!

Minor issues:

- Check accent on Nicol\`o Cesa-Bianchi in bibliography.
- Near the end of the proof of Theorem 6: I believe you mean Adam
  suffers a "regret" instead of a "loss" of at least 2C-4.
  Also 2C-4=2C-4 is trivial in the second but last display.

---

> ### Author Response · Authors · 2017-12-15
> **Thanks for the very helpful review**
>
> We deeply appreciate the reviewer for a thorough and constructive feedback.
>
> - Theorem 2 & 3 are much more involved and hence the aim of Theorem 1 was to provide a simplified counter-example for a restrictive setting, thereby providing the key ideas of the paper.
> - We will emphasize your point about projections in the final version of the paper.
> - We agree that the role of Theorem 4 right now is to provide a sanity check. Indeed, it is not possible to improve upon the of online gradient descent in the worst case convex settings. Algorithms such as Adagrad exploit structure in the problem such as sparsity to provide improved regret bounds. Theorem 4 provides some adaptivity to sparsity of gradients (but note that these are upper bounds and it is not clear if they are tight). Adaptive methods seem to perform well in few non-sparse and nonconvex settings too. It remains open to understand it in the nonconvex settings of our interest.
> - Indeed, there is a typo; we expect ||g{1:T,i}|| to grow like sqrt(T). The main benefit in adaptive methods comes in terms of sparsity (and dimension dependence). For example see Section 1.3 in Duchi et al. 2011). We have revised the paper to incorporate these changes.
> - We can indeed assume that x_1 = 1 (without loss of generality) because for any choice of initial point, we can always translate the function so that x_1 = 1 is the initial point in the new coordinate system. We will add a discussion about this in the final version of the paper.
> - The last part of Theorem 6 explains the reduction with respect to general epsilon. We will further highlight this in the final version of the paper.
>
> Other remarks:
>
> Regarding widespread adoption of Adam: It is possible that in certain applications the issues we raised in this work are not that severe (although they can still lead to degradation in generalization performance). On the contrary, there exist a large number of real-world applications, for instance training models with large output spaces, which suffer from the issues we have highlighted and non-convergence has been observed to occur more frequently. Often, this non-convergence is attributed to nonconvexity but our paper shows one of the causes that applies even to convex settings.
> As stated in the paper, using a problem specific large beta2 seems to help in some applications. Researchers have developed many tricks (such as gradient clipping) which might also play a role in mitigating these issues. We propose two different approaches to fix this issue and it will be interesting to investigate these approaches in various applications.
>
> We have addressed all other minor concerns directly in the revision of the paper.

---

### Official Review · AnonReviewer4 · 2017-12-14
**Theoretical (maybe practical) issues with ADAM, and an way to solve them**

**Rating:** 8
**Confidence:** 3

**Review:**

This paper examines the very popular and useful ADAM optimization algorithm, and locates a mistake in its proof of convergence (for convex problems). Not only that, the authors also show a specific toy convex problem on which ADAM fails to converge. Once the problem was identified to be the decrease in v_t (and increase in learning rate), they modified the algorithm to solve that problem. They then show the modified algorithm does indeed converge and show some experimental results comparing it to ADAM.

The paper is well written, interesting  and very important given the popularity of ADAM.

Remarks:
- The fact that your algorithm cannot increase the learning rate seems like a possible problem in practice. A large gradient at the first steps due to bad initialization can slow the rest of training. The experimental part is limited, as you state "preliminary", which is a unfortunate for a work with possibly an important practical implication. Considering how easy it is to run experiments with standard networks using open-source software, this can easily improve the paper. That being said, I understand that the focus of this work is theoretical and well deserves to be accepted based on the theoretical work.

- On page 14 the fourth inequality not is clear to me.

- On page 6 you talk about an alternative algorithm using smoothed gradients which you do not mention anywhere else and this isn't that clear (more then one way to smooth). A simple pseudo-code in the appendix would be welcome.

Minor remarks:
- After the proof of theorem 1 you jump to the proof of theorem 6 (which isn't in the paper) and then continue with theorem 2. It is a bit confusing.
- Page 16 at the bottom v_t= ... sum beta^{t-1-i}g_i should be g_i^2
- Page 19 second line, you switch between j&t and it is confusing. Better notation would help.
- The cifarnet uses LRN layer that isn't used anymore.

---

> ### Author Response · Authors · 2017-12-15
> **Thanks for the review**
>
> We thank the reviewer for the helpful and supportive feedback. The focus of the paper is to provide a principled understanding for the exponential moving average (EMA) adaptive optimization methods, which are now used as building blocks of many modern deep learning applications. The counter-example for non-convergence we show is very natural and is observed to arise in extremely sparse real-world problems (e.g., pertaining to problems with large output spaces). We provided two general directions to address the convergence issues in these algorithms (by either changing the structure of the algorithm or by gradually increasing beta2 as algorithm proceeds). We have provided preliminary experiments on a few commonly used networks & datasets but we do agree that a thorough empirical study will be very useful and is part of our future plan.
>
> - Fourth inequality on Page 14: We revised the paper to explain it further.
> - We will be happy to elaborate our comment about smoothed gradients in the final version of the paper.
> - We also addressed other minor suggestions.

---

### Public Comment · ~Mario_Ynocente_Castro1 · 2017-11-17
**Hyperparameters for experiments**

Hello,

I tried implementing AMSGrad (here: https://colab.research.google.com/notebook#fileId=1xXFAuHM2Ae-OmF5M8Cn9ypGCa_HHBgfG) for the experiment on the stochastic optimization setting and obtain that x_t approaches -1 faster that on the paper but convergence seems less stable, so I was wondering about the specific values for other hyperparameters like the learning rate and epsilon which weren't mentioned, in my case I chose a learning of 1e-3 and an epsilon of 1e-8 which seems to be the standard value on most frameworks.

---

> ### Author Response · Authors · 2017-11-20
> **Re: Hyperparameters for experiments**
>
> We thank you for your interest in our paper and for pointing out this missing detail. We use a decrease step size of alpha/sqrt(t) (as suggested by our theoretical analysis) for the stochastic optimization experiment. The use of decreasing step size leads to a more stable convergence to the optimal solution (especially in scenarios where the variance is reasonably high). We did not use epsilon in this particular experiment since the gradients are reasonably large (in other words, using a small epsilon like 1e-8 should produce more or less identical results). We will add these details in the next revision of our paper.

---

> > ### Public Comment · ~joe_denly1 · 2019-03-25
> > **developer**
> >
> > This is something which I am looking for and thanks for sharing the information with us. https://errorcode0x.com/error-code-0x80070002/  helped me to get the hyperparameters.

---

> > > ### Public Comment · ~Jon_Snow2 · 2019-09-14
> > > **developer**
> > >
> > > Can you guys help me to fix https://www.errorcodeshelp.com/0xc0000005/
> > > let me know if someone can help me here.

---

### Public Comment · ~David_Martínez1 · 2017-12-08
**Nice paper**

Congratulations for this paper, I really enjoyed it. It is a well written paper that contains an exhaustive set of counterexamples. I had also noticed that the proof of Adam was wrong and included it in my Master Thesis  (https://damaru2.github.io/convergence_analysis_hypergradient_descent/dissertation_hypergradients.pdf Section 2.4) and I enjoyed reading through the paper and finding that indeed it was not just that the proof was wrong but that the method does not converge in general, not even in the stochastic case.

I noticed some typos / minor things that seem that need to be fixed:

+ In the penultimate line of page 16 there is this equality v_{t-1} = .... g_i. This g_i should be squared.

+ In the following line, there is another square missing in a C, it should be (1-\beta_{t-1}_2)(C^2 p + (1-p)) and there is a pair of parenthesis missing in the next term, it should be  (1-\beta_2^{t-1})((1+\delta)C-\delta)

+ The fact that in Theorems 2 and 3 \beta_2 is allowed to be 1 is confusing, since the method is not well defined if \beta_2 is 1 (and you don't use an \epsilon in the denominator. If you use an \epsilon then with \beta_1 = 0 the method is equivalent to SGD so it converges for a choice of alpha). In particular, in the proof of theorem 3 \sqrt{1-\beta_2}  appears in some denominators and so does \sqrt{\beta_2} but there is no comment about what happens when this quantities are 0. There should be a quick comment on this or the \beta_2 \leq 1 should be removed from the theorems.

Best wishes

---

> ### Author Response · Authors · 2017-12-15
> **Thanks for the feedback**
>
> Thanks David, for your interest in this paper and helpful comments (and pointers). We have addressed your concerns regarding typos in the latest revision of the paper.

---

### Public Comment · (anonymous) · 2017-12-09
**Does similar analysis apply to the first moment of gradient? How about using a large beta1 and beta2?**

Thanks for the inspiring paper. The observations are interesting and important!

It is easy to capture that exponential moving average might not able to capture the long-term memory of the gradients.

The paper is mainly focused on the beta2 that involving the averaging of second moment. It makes me wonder whether the beta1 on the averaging of the first moment gradient also suffer the similar problem.

It seems a direct solution would be using a large beta1 and beta2. (Always keep the maximum of the entire history seems is not the best solution and an average over a recent history might be a better alternative.)

I did not carefully check the detail of the paper. But generally, one would have a similar concern I think. Could you explain the benefits of the proposed algorithm?

The synthetic experiments seem to use a relatively insufficient large beta2 regarding the large gradient gap, which makes it not able to capture the necessary long-term dependency.

---

> ### Author Response · Authors · 2017-12-15
> **Thanks for the feedback**
>
> Thanks for your interest in our paper and for your feedback. We believe that beta1 is not an issue for convergence of Adam (although our theoretical analysis assumes a decreasing beta1). For example, in stochastic convex optimization, momentum based methods have been shown to converge even for constant beta1. That said, it is indeed interesting to develop better understanding of the effect of momentum in convergence of these algorithms (especially in the nonconvex setting).
>
> As the paper shows, for any constant beta2, there exists a counter-example for non-convergence of Adam (both in online as well as stochastic setting, Theorem 2 & Theorem 3). Using a large beta2 can partially mitigate this issue in practice but it is not clear how high beta2 should be and it is indeed an interesting research question. Our paper proposes a couple of approaches (AMSGrad & AdamNC) for addressing these issues. AMSGrad allows us to use a fixed beta2 by changing the structure of the algorithm (and also allows us to use a much slow decaying learning rate than Adagrad). AdamNC looks at an approach where beta2 changes with t, ultimately converging to 1, hopefully allowing us to retain the benefits of Adam but at the same time circumventing its non-convergence.
>
> The aim of synthetic experiments was to demonstrate the effect of non-convergence. We can modify it to demonstrate similar problem for any constant beta2.

---

### Public Comment · (anonymous) · 2017-12-10
**Some questions about Theorem 2 and Corollary 1**

Dear authors,

It's a very good paper, but I have some questions as follows:

(1) In the last paragraph on Page 14, it says the fourth inequality is from $\beta^{i'-1}_2 C^2 \le 1$, but I couldn't go through from the third inequality to the fourth inequality on Page 14. It seems that you applied the lower bound of $v_{t+i-1}$ (i.e. $v_{t+i-1} \ge (1-\beta)\beta^{i-1}_2 C^2$ which is not desired) instead of its upper bound (which is truly required)?

(2) In Corollary 1, from my understanding, the L2 norm of $g_{1:T,1}$ should be upper bounded by $\sqrt(T)G_{\inf}$, so the regret be $O(\sqrt(T \logT))$ instead of $O(\sqrt(T))$ as stated in the remark of Corollary 1.

Correct me if I'm wrong. Thanks!

---

> ### Author Response · Authors · 2017-12-15
> **Thanks for the feedback**
>
> (1) Thanks for the interest in our paper and looking into the analysis carefully. We believe there is a misunderstanding regarding the proof. The third inequality follows from the lower bound v_{t+i-1} \ge (1-\beta)\beta^{i-1}_2 C^2. The fourth inequality actually follows from the upper bound on v_{t+i-1}  (which implicitly uses \beta^{i'-1}_2 C^2 \le 1). We revised the paper to provide the detailed derivation, including specifying precise constants that were previously omitted.
>
> (2) Actually, an easy observation from our analysis is that we can bound the regret of AMSGrad by O(G_infty sqrt(T)) as well. This can be easily seen from the proof of Lemma 2 where in the analysis the term \sum_{t=1}^T |g_{t,i}/\sqrt{t} can be bounded by O(G_infty sqrt(T)) instead of O(\sqrt{\log(T) ||g_{1:T}||_2). Thus, the regret of AMSGrad is upper bounded by minimum of O(G_infty sqrt(T)) and the bound presented in Theorem 4, and thus the worst case dependence on T is \sqrt{T} rather than \sqrt{T \log(T)}. We will make this point in the final version of the paper.

---

### Public Comment · (anonymous) · 2018-03-08
**one question on lemma 2 to prove theorem 4**

Hello authors, it is a great work. However, I have a question on the first equation (the second inequality sign) of the proof of lemma2. This equation said:

\sum_{t=1}^T \alpha_t || \hat V_t^{-1/4}  m_t ||^2 \leq  \sum_{t=1}^{T-1} \alpha_t || \hat V_t^{-1/4}  m_t ||^2 + \alpha \sum_{i=1} ( numerator / denominator )

My question is on this numerator (m_{T,i}^2) which you applied the update m_t = \beta_{1t} m_{t-1} + (1-\beta_{1t}) g_t. Since in your proof beta_{1t} is non-increasing (changes) over time, I wonder if the numerator in your equation is correct. It looks that the coefficient (1-\beta_{1j}) before g_{j,i} is ignored. If \beta_{1j} is a constant \beta, then the derived bound is correct since (1-beta)^2 \leq 1. However, the effect of varying \beta_{1j} makes the derivation involved. Could you please elaborate on why the term (1-\beta_{1j}) can be ignored, or how you derive the numerator in the paper?

Thanks,

---

> ### Author Response · Authors · 2018-05-31
> **beta_{1t} is upper bounded**
>
> Thanks for your interest in the paper. This is obtained by applying Cauchy-Schwarz inequality and using the fact that 1 - \beta_{1t} is bounded by 1.

---

### Public Comment · ~Jeremy_Bernstein1 · 2018-04-25
**Is the problem with Adam, or with the theoretical framework used to analyse it?**

Dear authors,

Thank you for the valuable contribution in pointing out issues in the original Adam convergence proof, and noticing that bimodal noise distributions can obstruct convergence.

In the paper you suggest modifying the Adam algorithm to solve the problem. I want to add two much simpler fixes that do not involve modifying Adam at all:

1. A cheap trick: set epsilon large.
----for large enough epsilon, Adam becomes exactly SGD.
----the convergence of SGD is well-established.
----how large should epsilon be? Well if you don't know in advance, you can use a growing schedule.

2. A more subtle fix: use a larger minibatch size.
----your construction relies on bimodal gradient noise
----but by the central limit theorem, even for modest mini-batch sizes the stochastic gradient noise will become nice and Gaussian
----this issue is studied in greater depth in our paper [1]
----we establish convergence for Adam with beta1=beta2=epsilon=0, otherwise known as signSGD

These observations are important because I suspect that AMSgrad, similar to Adagrad, will not scale up to deeper networks, whereas Adam, as we all know, does.

On a meta-level, the problem you raise is an issue with the analytical framework, rather than with the Adam algorithm itself. In particular:
----the analytical framework provides no knob to tune the mini-batch size
----in your Theorem 6 you assume epsilon is chosen before seeing the optimisation problem. But in real life we can choose our hyperparameters after seeing the optimisation problem.

With regard to this comment in the abstract:
>> "In many applications, e.g. learning with large output spaces, it has been empirically observed that these algorithms fail to converge to an optimal solution (or a critical point in nonconvex settings)."

My claim is that some practitioners have problems with Adam not because it has fundamental convergence issues but because it has too many hyperparameters for them to tune carefully, especially when they do not know what each hyperparameter is actually controlling. For any problem, a properly tuned-Adam will converge at least as well as SGD and signSGD **since it contains these algorithms as special cases**. The practical takeaway: the practitioner's dream of having magic hyperparameters that work for every problem under the sun is exactly that, a dream :)

[1] signSGD: compressed optimisation for non-convex problems (https://arxiv.org/abs/1802.04434)

---

> ### Author Response · Authors · 2018-05-31
> **TL;DR : Its with the algorithm**
>
> Dear Jeremy,
>
> Thank you for your interest in the paper.
>
> To answer your question "Is the problem with Adam ....?" : Our paper shows that the algorithm defined in the Adam paper  (https://arxiv.org/pdf/1412.6980.pdf, Algorithm 1)  (including one with decreasing step size alpha) has convergence issues. Specifically, for any setting of the Adam parameters (beta_1, beta_2, minibatch size, epsilon, etc) there is a convex optimization setting where Adam will not converge to the optimal solution, even if decreasing learning rates are used. This is in contrast to algorithms like SGD which, with decreasing learning rates, is guaranteed to converge.
>
> As to how to fix the convergence issues, there are quite a few ways and our paper mentions a couple of them. Using a high epsilon as mentioned in the paper (please refer to the discussion after Theorem 3 in our paper), or using increasing schedule for epsilon is another fix. Also, as also studied in https://arxiv.org/pdf/1507.02030.pdf and probably other papers, increasing minibatch can also help in the stochastic setting. But it is important to understand the practical implications of any of these fixes because they involve adding more hyper-parameters, schedules and may be more resources. We proposed two solutions to tackle the aforementioned issues, including one which does not have any additional parameters (AMSGrad).
>
> Of course, Adam (Algorithm 1) in https://arxiv.org/pdf/1412.6980.pdf roughly subsumes SGD by using a large value of epsilon and obviously one should simply use SGD instead of Adam with very large (and increasing with iterations) epsilon if the former works well. But several question remain: whether simply using SGD is better or are there other values (or schedules) of Adam that make it work better (and furthermore, is it dimensionality dependent?). Or is there a modification of the algorithm that can tackle these issues with fewer hyper-parameters? While designing any of these algorithms, it is important to not add too many additional hyper-parameters because tuning them can be very difficult in practice.

---

### Public Comment · ~Martin_Georg_Weiß1 · 2018-04-27
**Are the counter examples valid?**

Your change in the algorithm using the maximum is a great progress. Two of my master students also noticed that there are several errors in the proof of Kingma and Ba. They corrected some constants and parts of the proof, but the main points seem beyond repair. Because both master theses contain confidential material from industrial applications only some small part has been published, see https://www.researchgate.net/publication/324808725_An_improvement_of_the_convergence_proof_of_the_ADAM-Optimizer

However the counter examples in your paper do not seem valid to me: The task to be solved is not clearly defined in your paper, neither in Kingma and Ba. The task should be to minimize the expectation of some loss function L(x,z) like a the squared error between network output and training sample, with x the parameters (network weights) and z the sample from the same probability distribution for all t. So the the loss function usually is the same for all t, and t appears in f_t(x) = L(x,z_t) only via the sample z_t. The random variables do not know what time t it is, only the algorithm does. So your counterexample cannot decide what t mod 3 is. If your assumption is that the probability changes periodically with t that should be stated. The same reasoning holds for minimizing a deterministic function f(x), then dependence on t is only via f(x_t) as well.

You might argue: One can define a (probably very complicated and spiky) function g(x) not depending on t which gives the same function values for all x_t generated by the counterexample. But then it is not clear whether such a  function exists: If e.g. t1 mod 3 = 1and t2 mod 3 = 2 holds, but at the same time x_t1 = x_t2, one cannot assign a unique function value for g(x_t1) .
The implication is: The Adam convergence proof is wrong, but the counter examples are not relevant. So there might still be hope for a convergence proof for Adam, maybe under additional assumptions, but without the maximum.

Could you please comment on that.

---

> ### Author Response · Authors · 2018-05-31
> **Problem setting in the paper**
>
> Dear Martin,
>
> Thanks for your interest in our paper. We think there is a misunderstanding regarding the online setting. Please refer to Section 2 of the paper. Also, please refer to http://www.jmlr.org/papers/volume12/duchi11a/duchi11a.pdf and references therein for a more detailed description of the setting. Our results also apply to the stochastic setting (see Theorem 3 in the paper).

---

### Public Comment · ~Daniel_Povey1 · 2018-07-27
**Using previous t's 2nd moment estimate**

Guys,
I know it's been a while since this conversation was active, but I just noticed this paper and thread, and wanted to comment.

To me the obvious problem with Adam that would prevent convergence is that it uses the 2nd moment estimate from the *current* t, instead of the *previous* t.   That introduces a nonlinearity in the dependence on the currently selected sample, which I think would tend to be fatal for most proofs.

Your approach seems to be to try to remove the dependence on the current sample by making the history very long.

---

> ### Public Comment · ~Daniel_Povey1 · 2018-07-27
> **oh, wait...**
>
> sorry.. on reviewing how adam works again, I see that the first moment, as well as the 2nd moment, are moving averages, so changing t to t-1 wouldn't change the fact that you have the same samples appear in both the numerator and denominator of the parameter-change equation delta-params = moving-average-mean / sqrt(moving-average-variance + epsilon).
> To fix it, I guess you could either use a fix like you are proposing, involving taking the average over more samples, or you could take steps to make sure that the samples appearing in the numerator and denominator of the equation were independent, e.g. using some odd-even formulation to separate the two moving averages into two bins and crossing them over so that bin A's mean was divided by bin B's variance  and vice versa.
> To me the easiest demonstration of the problem is a parameter whose derivative at equilibrium is small and negative for almost all samples, and large and positive on very rare samples.  Because you divide by that large number whenever you encounter those rare samples, the expected parameter change is negative in a situation where it should be zero.

---

> ### Public Comment · (anonymous) · 2019-04-27
> **A related paper.**
>
> Hi, there is a paper "AdaShift: Decorrelation and Convergence of Adaptive Learning Rate Methods" which solves the non-convergence of Adam via using temporally decorrelated g_t and shares the spirit with your comment. See https://openreview.net/forum?id=HkgTkhRcKQ for more details.

---

### Public Comment · ~Router_Error_Code1 · 2019-12-30
**It,s Really Hepful**

I was looking for router error code 488

---

### Public Comment · ~Joseph_Diaz1 · 2020-03-15
**I have add multiple Solutions to fix**

Here is a link to the solution: https://magicvibes.co/fix-0x00000002-error/

---

### Decision · Program_Chairs · 2018-01-29
**ICLR 2018 Conference Acceptance Decision**

**Decision:**

Accept (Oral)

**Comment:**

This paper analyzes a problem with the convergence of Adam, and presents a solution. It identifies an error in the convergence proof of Adam (which also applies to related methods such as RMSProp) and gives a simple example where it fails to converge. The paper then repairs the algorithm in a way that guarantees convergence without introducing much computational or memory overhead. There ought to be a lot of interest in this paper: Adam is a widely used algorithm, but sometimes underperforms SGD on certain problems, and this could be part of the explanation. The fix is both principled and practical. Overall, this is a strong paper, and I recommend acceptance.